# Solving Challenging Math Word Problems Using GPT-4 Code Interpreter with Code-Based Self-Verification

**Aojun Zhou[1][*]   Ke Wang[1][*]   Zimu Lu[1][*]   Weikang Shi[1][*]   Sichun Luo[3][*]   Zipeng Qin[1]**
**Shaoqing Lu[4]   Anya Jia[5]   Linqi Song[3]   Mingjie Zhan[1][†‡]   Hongsheng Li[1,2][‡]**

[1]MMLab, The Chinese University of Hong Kong  [2]Shanghai Artificial Intelligence Laboratory
[3]City University of Hong Kong          [4]CSUST          [5]Tufts University
{aojunzhou, zmjdll}@gmail.com   hsli@ee.cuhk.edu.hk

## Abstract

Recent progress in large language models (LLMs) like GPT-4 and PaLM-2 has brought significant advancements in solving math problems. In particular, OpenAI's latest version of GPT-4, known as GPT-4 Code Interpreter, shows remarkable performance on challenging math datasets. In this paper, we explore the effect of code on enhancing LLMs' reasoning capability by introducing different constraints on the *Code Usage Frequency* of GPT-4 Code Interpreter. We found that its success can be primarily attributed to its powerful skills in generating and executing code, evaluating the execution result, and rectifying its solution when receiving unreasonable outputs. Based on this, we propose a novel prompting method, explicit code-based self-verification (CSV). This method employs a zero-shot prompt on the GPT-4 Code Interpreter to encourage it to use code to self-verify its answers. In instances where the verification state is "False", the model will automatically amend its solution. Furthermore, we recognize that the states of the verification result indicate the confidence of a solution, which can improve the effectiveness of majority voting. With GPT-4 Code Interpreter and CSV, we achieve an impressive zero-shot accuracy of various mathematical problem-solving benchmarks.

## 1 Introduction

Large language models (LLMs) (Brown et al., 2020; OpenAI, 2023; Anil et al., 2023) have shown impressive success in various tasks. However, they still fall short in mathematical reasoning, often producing nonsensical or inaccurate content and struggling with complex calculations. Previous works to tackle these challenges include the Chain-of-Thought (CoT) (Wei et al., 2022) framework, which enhances LLMs' logical reasoning abilities by generating intermediate reasoning steps. Additionally, PAL (Gao et al., 2023) uses the Python interpreter to improve computational accuracy.

Recently, OpenAI has unveiled an improved version of GPT-4, namely the GPT-4 Code Interpreter[1] or *GPT4-Code*, which is good at providing natural language reasoning, alongside step-by-step Python code. Notably, it can generate and execute code incrementally, and subsequently present the execution results back to the LLM. This mechanism has shown promising results in solving mathematical problems. Our initial experiments show that GPT4-Code achieved an impressive zero-shot accuracy of 69.69% on the challenging MATH dataset (Hendrycks et al., 2021), marking a significant improvement of 27.5% over GPT-4's performance (42.2%).

While GPT4-Code has demonstrated proficiency in solving math problems, there has been a notable absence of systematic analysis focusing on understanding and further enhancing its mathematical problem-solving abilities. A critical distinction between GPT4-Code and its predecessor, GPT-4, lies in GPT4-Code's ability to *automatically generate and execute code*. Therefore, this paper

---

[*]Equal contribution.

[†]Project leader.

[‡]Corresponding author.

[1]https://chat.openai.com/?model=GPT4-Code-interpreter

presents pilot experiments investigating GPT4-Code's code generation and execution mechanism using specific code-constrained prompts. The analysis reveals that GPT4-Code's strong performance is not solely due to its code generation and execution abilities but also its capacity to adjust its problem-solving strategies based on feedback from code execution—a process we term **self-debugging** (akin to (Chen et al., 2023b)), examples illustrated in Appendix E. Due to this clever mechanism, there is an increased frequency of code usage. Hence, we introduce Code Usage Frequency to differentiate these unique prompting strategies to quantitatively analyze the impact of code-constrained prompts on GPT4-Code for mathematical problem-solving. The step-by-step code generation and self-debugging mechanisms highlight the critical role of code in mathematical problem-solving. Nevertheless, the self-debugging mechanism only verifies the correctness of code while lacking the verification of the reasoning steps and the final answer, which has been demonstrated to be of vital importance to solve math problems of LLMs (Cobbe et al., 2021; Lightman et al., 2023).

We therefore ask the question: *can we fully exploit the code generation and self-debugging mechanisms in GPT4-code, so that it can automatically verify and correct its solutions, without extra assistance from other models or users?*

To answer this question, we propose a simple yet effective prompting technique termed the **explicit code-based self-verification** (CSV), which guides GPT4-Code to generate additional code that verifies the answer and adjusts the reasoning steps if there's a flaw in reasoning. Unlike previous methods that rely on external language models for verification (Lightman et al., 2023; Cobbe et al., 2021), our approach leverages GPT4-Code's inherent strengths. This approach offers two key benefits: (1) When the verification indicates an answer is **False**, GPT4-Code can rectify its prior solution and provide an improved alternative. (2) Solutions verified as **True** tend to be more reliable, akin to human problem-solving. However, even if a solution is self-verified as **False**, we do not directly abandon it. Instead, we propose a weighted majority voting strategy that incorporates the code-based solution verification results, as opposed to relying exclusively on the frequency of answers. We assign different weights to the solutions according to their verification states, reflecting the solutions' varying levels of reliability. In alignment with the Code Usage Frequency analysis from our pilot experiments, our explicit code-based self-verification prompt boosts GPT4-Code's accuracy in mathematical problem-solving with increased code usage.

Empirical study demonstrates the effectiveness of our proposed pipeline on the MATH, GSM8K, and MMLU-Math datasets using GPT4-Code. Our method achieves an impressive accuracy of **84.3%** on the MATH dataset, greatly outperforming the **base GPT4-Code** and previous SOTA methods.

This paper's main contributions can be summarized in three key aspects:

- This study provides the first systematic analysis of code generation, execution, and self-debugging's role in mathematical problem-solving. Our findings reveal that GPT4-Code's impressive mathematical problem-solving proficiency is primarily attributed to its step-by-step code generation and dynamic solution refinement based on code execution outcomes.

- We introduce the innovative *explicit code-based self-verification (CSV) prompt*, which leverages GPT4-Code's advanced code generation mechanism. This prompt guides the model to verify the answer and then reevaluate its solution with code. CSV not only extends the verification to the logic behind problem-solving but also improves the efficacy of the majority voting method by integrating the verification states.

- We have demonstrated the superior performance of GPT4-Code and the proposed CSV pipeline through extensive *quantitative experiments* on a variety of challenging mathematical reasoning datasets.

## 2 RELATED WORK

**Chain-of-Thought Reasoning.** The *Chain-of-Thought (CoT)* prompting approach proposed by (Wei et al., 2022) shows the multi-step reasoning capabilities of LLMs. (Kojima et al., 2022) implements *Zero-shot-CoT*, which can serve as a strong zero-shot baseline. Further research extends the reasoning capabilities of CoT by applying majority voting (Wang et al., 2023), choosing complex few-shot examples and outputting results (Fu et al., 2022), breaking down the problem (Zhou et al., 2023), or even using Tree-of-Thoughts (Yao et al., 2023). Similar to Zero-shot-CoT, our method applies "step by step"-like prompts to regularize GPT4-Code's use of code. Additionally, We enhance majority voting by leveraging the results of CSV as voting weights.

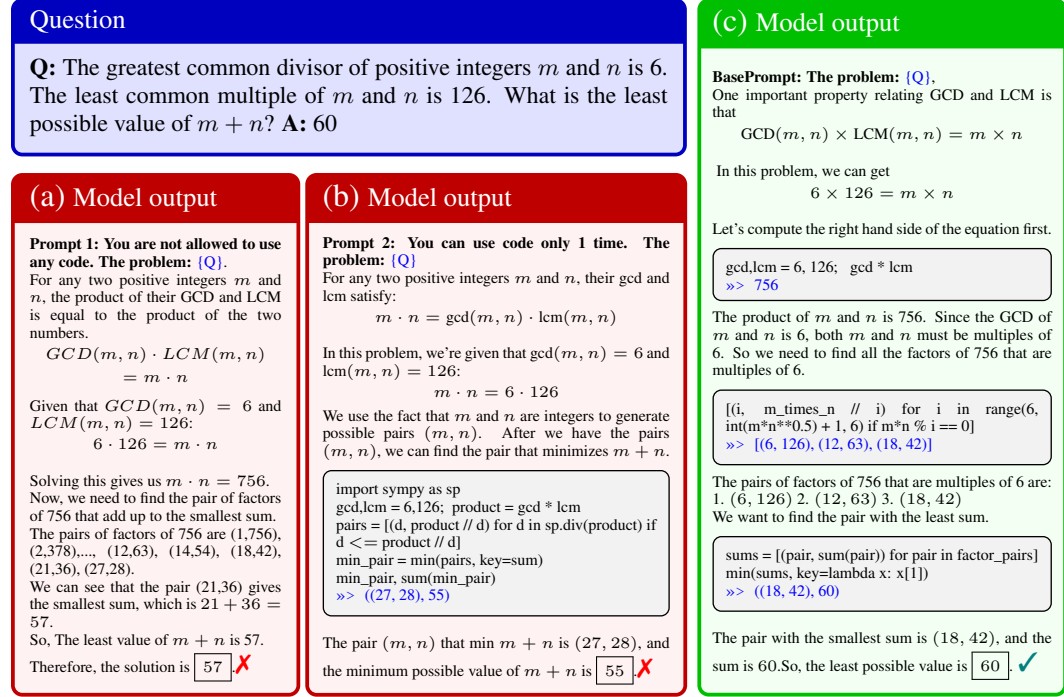

**Figure 1:** Example from the 277th number theory problem in the MATH dataset: **(a)** Prompt 1 generates text-only reasoning steps. **(b)** Prompt 2 generates intermediate text reasoning steps and post-hoc code, akin to PAL (Gao et al., 2023). **(c)** Base prompt without code usage constrain. Details of the three prompts are presented in Sec. 3.1. (For more examples, see Tab. 10 and Tab. 11 in Appendix H.)

**Solving Math Problems with Code.** Large language models have been found to be less accurate in performing arithmetic calculations (Cobbe et al., 2021; Lewkowycz et al., 2022; Gao et al., 2023; Lu et al., 2022). Consequently, previous works have attempted to solve math problems with the assistance of code. The GSM8K dataset (Cobbe et al., 2021) uses calculation annotations to extract all arithmetic calculations solved by an external calculator: the Python *eval* function. Program-Aided Language model (PAL) (Gao et al., 2023) and Program of Thoughts (PoT) (Chen et al., 2022) obtain the answer by generating and executing Python code. Although they can improve computational accuracy, many generated codes get wrong answers due to the lack of verification. Our approach not only utilizes the ability of GPT4-Code to generate codes and refine codes that fail to run, but also uses CSV to enhance the accuracy of the answers.

**Self-Verification.** Previous studies train an additional verifier to verify the correctness of final answers (Cobbe et al., 2021) or intermediate steps (Lightman et al., 2023; Li et al., 2023). (Weng et al., 2023) showed the self-verification abilities of LLMs by generating and ranking multiple answers. Furthermore, *Self-refine* proposed by (Madaan et al., 2023) iteratively refines its output through self-generated feedback. *Self-debug* (Chen et al., 2023b) prompts the LLM to debug its own prediction for code generation. Unlike these methods that require LLMs to give verification feedback in natural language, our method applies generated codes to verify the answers and votes on different answers based on the verification results, thus improving the accuracy of the verification.

## 3 METHOD

We first conduct a pilot experiment with GPT4-Code on the challenging MATH dataset (Hendrycks et al., 2021). Remarkably, it achieves an accuracy of 69.69%, significantly surpassing the previous state-of-the-art performance of 53.9% (Zheng et al., 2023). Encouraged by the compelling performance of GPT4-Code, we strive to systematically explore and analyze its underlying code mechanisms. In Sec. 3.1, we illustrate, via our code-constrained prompts design, that GPT4-Code's robust performance in solving math problems derives not only from its ability to generate accurate step-by-step code, but also from its self-debugging mechanism. In Sec. 3.2, we aim to leverage GPT4-Code's self-debugging strengths to further improve its mathematical problem-solving ability.

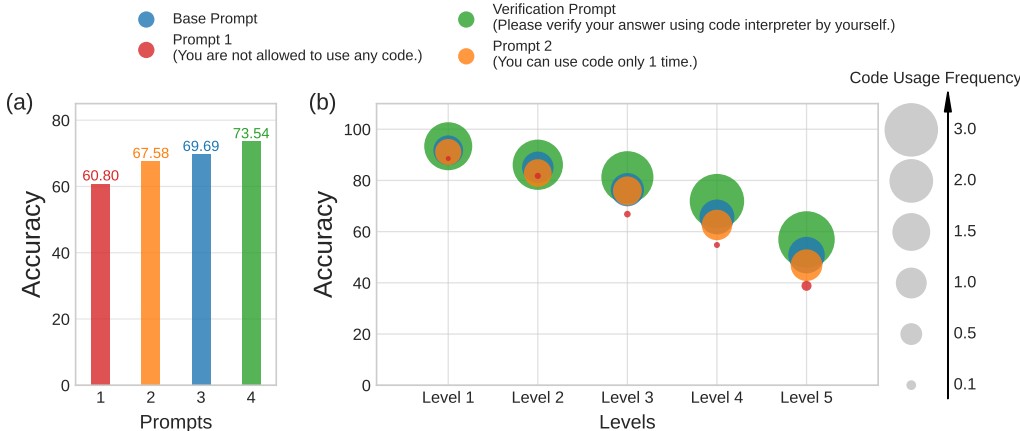

**Figure 2:** Performance on MATH dataset of different levels by applying different prompts to adjust the frequency of code usage. **(a)** Comparison of overall accuracy between the four prompts. **(b)** Code Usage Frequency is in proportion to accuracy in all five levels, and this phenomenon is especially apparent when the problems are relatively complicated (i.e., with higher levels). The red points denoting Prompt 1 show that the model still occasionally uses code, especially when the problem is very difficult. However, even then, the Code Usage Frequency is negligible.

### 3.1 Pilot Experiments on analyzing Code Usage of GPT4-Code

To explore the impact of code on GPT4-Code's mathematical skills, we adopt a straightforward approach by constraining GPT4-Code's uasge of code through thoughtfully constructed prompts. Specifically, we introduce two code-constrained prompts and a base prompt for comparison:

- **Prompt 1.** *No code usage is allowed*: With this prompt, GPT4-Code is prohibited from using code. This prompts GPT4-Code to rely solely on Natural Language (**NL**) reasoning chain, resembling solutions in the CoT framework (Wei et al., 2022). The resulting sequence of reasoning steps is depicted as $\mathbf{C_{NL}}$, with an example given in Fig. 1 (a).

- **Prompt 2.** *Code can be used only once*: In this prompt setting, GPT4-Code is permitted to employ code within a single code block to generate the solution, mirroring the PAL approach introduced by (Gao et al., 2023). We denote this sequence as $\mathbf{C_{SL}}$, representing a series of Symbolic Language (**SL**), such as Python. An example is shown in Fig. 1 (b).

- **Base Prompt.** GPT4-Code is prompted to tackle the problem without any restrictions on code usage. This prompt leads to GPT4-Code's usual performance, which can be denoted as $\mathbf{C} = ((\mathbf{c1_{NL}}, \mathbf{c1_{SL}}), (\mathbf{c2_{NL}}, \mathbf{c2_{SL}}), \dots)$, representing a list of reasoning steps, each consisted of both natural language and code, with an example shown in Fig. 1 (c).

Apart from the specific example in Fig. 1, we introduce *Code Usage Frequency* to record the number of code executions for different prompts. The results of the experiments using these prompts are shown in Fig. 2 (b). This figure illustrates a positive correlation between the better performance of GPT4-Code and the higher Code Usage Frequency. More specifically,

**Prompt 1** v.s. **Prompt 2.** *Prompt 1* results in almost negligible code usage, while *Prompt 2* results in approximately 1 time's code usage. *Prompt 2* yields an accuracy gain of 6.78 percent over Prompt 1. This suggests that the Python code chains $\mathbf{C_{SL}}$ can improve computational capability more than the natural language chains $\mathbf{C_{NL}}$. This observation is consistent with the findings in previous Python-based prompting methods (Gao et al., 2023; Chen et al., 2022). However, employing code only once comes with an inherent drawback – the model lacks the ability to self-debug when the code output triggers an error or produces an implausible outcome.

**Prompt 2** v.s. **Base Prompt.** The *Base Prompt* consistently produces solutions that entail multiple instances of code usage, resulting in a large Code Usage Frequency. Additionally, the Base Prompt's accuracy of 69.69% represents a 2.11% improvement over Prompt 2's 67.58%. These improvements in Code Usage Frequency and accuracy might be attributable to two unique advantages: (1) Generating code in brief and frequent segments, divided among natural language reasoning steps, tends to result in higher accuracy. (2) The model possesses the capability to evaluate the results of

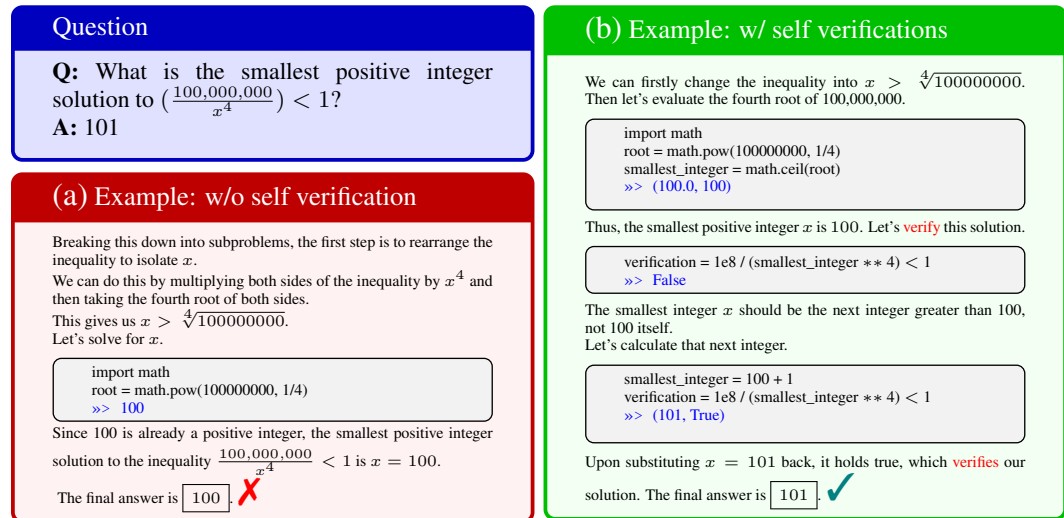

**Figure 3:** Question from the 712th intermediate algebra problem in the MATH dataset. **(a)** Without self-verification, the model generates a wrong answer. **(b)** With self-verification, the model corrects the error and generates the correct answer. **The CSV prompt**: *Solve the problem using code interpreter step by step, even in every sub-step. And following your answer, please verify it using code interpreter by yourself.*

code execution and make corrections to solution steps if the outcomes contain bugs or are deemed illogical, as illustrated in Tab. 8 and Tab. 9 (Appendix E).

From these observations, it is plausible to enhance and build upon the favorable attributes of GPT4-Code to further improve its precision in tackling math problems.

## 3.2 EXPLICIT CODE-BASED SELF-VERIFICATION PROMPTING

Inspired by the observations on Code Usage Frequency analysis, we seek to harness the capabilities of GPT4-Code. These capabilities include the model's aptitude for generating accurate code, evaluating the outcomes of code execution, and automatically adjusting reasoning steps of solutions when needed. Our objective is to utilize these strengths to augment solution verification.

To achieve this objective, we propose the technique termed as **explicit code-based self-verification** (CSV). This method prompts GPT4-Code to explicitly validate its answer through code generation. By implementing this prompt, we introduce an extra verification stage to the solution $\mathbf{C}$, referred to as $\mathbf{V}$. The verification result $\mathbf{V}$ can be classified as *True*, *False*, or *Uncertain*. An *Uncertain* classification indicates that GPT4-Code encountered difficulties in identifying an effective method for answer verification, thereby abstaining from delivering a definitive verification result. Leveraging GPT4-Code's inherent autonomous capabilities, we can formulate the proposed prompting as:

$$\mathbf{C} \to \mathbf{V} = \begin{cases} \text{True} & \to \text{final answer} \\ \text{False} & \to \mathbf{C}_{\text{new}} \to \mathbf{V} \to \cdots \to \text{True} \to \text{final answer} \\ \text{Uncertain} & \to \text{final answer} \end{cases}$$

An example is presented in Fig. 3 (b). Incorporated with CSV, the model becomes capable of using code to verify answers, then reviewing and adjusting how it arrived at the solution if the verification result is *False*, aiming at obtaining the correct answer. The different types of verification code can be seen in Tab.16, Tab.17, Tab.18, and Tab.19 (Appendix J). Upon refining and correcting the initial solution, we anticipate a notable increase in accuracy. It is worth noting that both the verification and rectification stages are code-based. This inevitably results in increased Code Usage Frequency, akin to the aforementioned analysis, which will be further demonstrated in subsequent experiments.

We perform experiments with CSV, and these results can be found in Fig. 2. The experiment here is conducted with GPT4-Code on MATH (Hendrycks et al., 2021). In Fig. 2 (b), the accuracy achieved

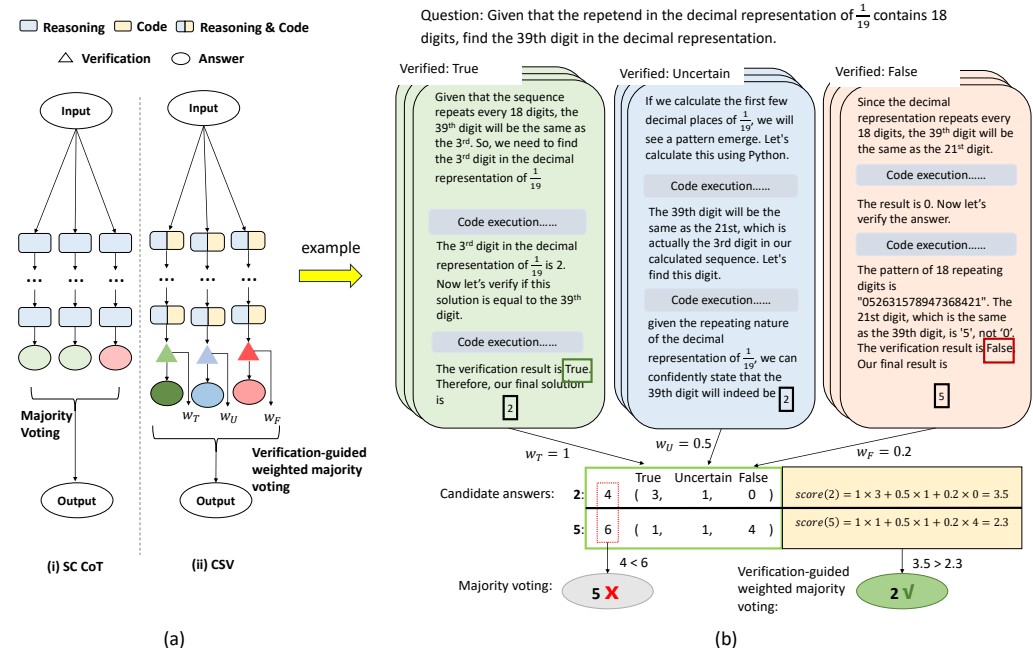

**Figure 4:** **(a)** Illustration of the Naive majority voting (Wang et al., 2023) and our Verification-guided weighted majority voting. **(b)** The full pipeline of the proposed Verification-guided Weighted Majority Voting (VW-voting) framework. We detect the self-verification state of each solution and classify them into three states: *True*, *Uncertain*, and *False*. According to the state of the verification, we assign each solution a different weight and use the classified result to vote the score of each possible answer. (For more examples, see Tab. 12 and Tab. 13 in Appendix G.)

with our proposed CSV prompt consistently surpasses that of the *Base Prompt* across all designated difficulty levels[2]. Meanwhile, the Code Usage Frequency receives a clear increase.

Before the advent of GPT4-Code, prior frameworks (Lightman et al., 2023; Cobbe et al., 2021) relied on an external Large Language Model (LLM) and well-constructed few-shot prompts for natural language verification. In contrast, GPT4-Code's robust capabilities enable our approach to depend solely on a straightforward prompt, thereby operating in a zero-shot manner. This enables GPT4-Code to autonomously verify and independently rectify its solutions using the advanced code execution mechanism, thereby eliminating the need for customized few-shot examples.

Given that CSV can effectively verify problem-solving answers, we can naturally integrate the verification states into majority voting, akin to the methodology embraced in self-consistency CoT (Wang et al., 2023). Answers deemed **True** through verification are generally more trustworthy, reflecting the problem-solving approach seen in human cognition (Newell & Simon, 1972; Wang & Chiew, 2010). This improved reliability can be leveraged in the widely-used majority voting process. To exploit this insight, we introduce *verification-guided weighted majority voting*, which assigns different weights to the states of the verification process.

In practice, it sometimes occurs that once an answer is confirmed as False, no additional verification is conducted, yielding a False verification state. We allocate corresponding weights these states of **True**, **Uncertain**, **False**: $w_{\mathrm{T}}, w_{\mathrm{U}}$, and $w_{\mathrm{F}}$, respectively.

Similar to the Self-consistency with CoT (CoT-SC) (Wang et al., 2023) in Fig. 4 (a)(ii), our framework can sample $k$ paths. For simplicity, we extract pairs of final answers and their corresponding verification results from $k$ solutions, represented as $(v^i, a^i), i = 1, 2, \ldots, k$, where $v^i$ and $a^i$ denote the $i$-th final answer and final verification result, respectively.

So the voting score for each candidate answer $a$ can be expressed as:

$$\text{Score}(a) = \sum_{\{v^i\}} w_v(\#\{i \mid a^i = a \text{ and } v^i = v\}), \quad v \in \{\text{True}, \text{Uncertain}, \text{False}\}, \tag{1}$$

---

[2]Human-perceived easier problems are categorized under Level-1 difficulty as per (Hendrycks et al., 2021).

**Table 1:** Accuracy (%) on MATH dataset. **VW-voting** is an abbreviation for Verification-guided Weighted Majority Voting. **Voting** is an abbreviation for Naive Majority Voting. (**Overall:** The results across various MATH subtopics)

| | Code-based Verification | Intermediate Algebra | Precalculus – | Geometry – | Number Theory | Counting & Probability | PreAlgebra – | Algebra – | Overall MATH |
|---|---|---|---|---|---|---|---|---|---|
| GPT-4 (OpenAI, 2023) | ✗ | - | - | - | - | - | - | - | 42.20 |
| GPT-3.5(CoT) (Zheng et al., 2023) | ✗ | 14.6 | 16.8 | 22.3 | 33.4 | 29.7 | 53.8 | 49.1 | 34.12 |
| GPT-4 (Complex CoT) (Fu et al., 2022) | ✗ | 23.4 | 26.7 | 36.5 | 49.6 | 53.1 | 71.6 | 70.8 | 50.36 |
| GPT-4 (PHP) (Zheng et al., 2023) | ✗ | 26.3 | 29.8 | 41.9 | 55.7 | 56.3 | 73.8 | 74.3 | 53.90 |
| GPT4-Code (**baseline**) | ✗ | 50.1 | 51.5 | 53.4 | 77.2 | 70.6 | 86.3 | 83.6 | 69.69 |
| GPT4-Code + **CSV** | ✓ | 56.6 | 53.9 | 54.0 | 85.6 | 77.3 | 86.5 | 86.9 | 73.54 |
| *Improvement* | | **+6.5** | **+2.4** | **+0.6** | **+8.4** | **+6.7** | **+0.2** | **+3.3** | **+3.85** |
| GPT4-Code + **Voting (k=16, baseline)** | ✗ | 63.3 | 64.1 | 61.7 | 89.1 | 84.6 | 90.8 | 92.9 | 79.88 |
| GPT4-Code + **CSV + Voting (k=16)** | ✓ | 72.7 | 66.5 | 64.5 | 93.1 | 88.8 | 91.2 | 95.3 | 83.54 |
| *Improvement* | | **+9.4** | **+2.4** | **+2.8** | **+4.0** | **+4.2** | **+0.4** | **+2.4** | **+3.66** |
| GPT4-Code + **CSV + VW-Voting (k=16)** | ✓ | **74.4** | **67.8** | **64.9** | **94.1** | **89.0** | **91.6** | **95.6** | **84.32** |
| *Improvement* | | **+11.1** | **+3.7** | **+3.2** | **+5.0** | **+4.4** | **+0.8** | **+2.7** | **+4.44** |

Here, $a$ represents a candidate answer, $v$ denotes the state of verification, and $w_v$ is an element from the set $\{w_{\mathrm{T}}, w_{\mathrm{U}}, w_{\mathrm{F}}\}$. Each $w_v$ signifies the degree of confidence associated with its corresponding verification state. Finally, we select the answer with the highest score from all candidate answers.

It should be noted that when $w_v = 1$ for all $w_v \in \{w_{\mathrm{T}}, w_{\mathrm{U}}, w_{\mathrm{F}}\}$, Eq. 1 becomes equivalent to the naive majority voting employed in Self-Consistency with CoT (CoT-SC) (Wang et al., 2023). Typically, we set $w_{\mathrm{T}} > w_{\mathrm{U}} > w_{\mathrm{F}}$, which means that an answer verified true has greater confidence than the one with an uncertain state of verification, while an answer verified false has the lowest degree of confidence. An example of the calculation process within verification-guided weighted majority voting is illustrated in Fig. 4.

## 4 EXPERIMENTS

**Datasets and Baseline.** We evaluate GPT4-Code using CSV on three datasets: MATH (Hendrycks et al., 2021), GSM8K (Cobbe et al., 2021), and MMLU-Math (Hendrycks et al., 2020). We primarily compare our code-based self-verification (CSV) approach to standard zero-shot prompting using GPT4-Code to validate the effectiveness of the self-verification ability. To more comprehensively evaluate the zero-shot capabilities of both GPT4-Code and GPT4-Code with CSV in mathematical reasoning tasks, we also compare our method with the state-of-the-art few-shot in-context-learning method using GPT-4 from PHP (Zheng et al., 2023) and Model selection (Zhao et al., 2023).

**Prompt.** The proposed prompt is presented in the caption of Fig. 3.

### 4.1 PERFORMANCE ON MATH

The MATH dataset (Hendrycks et al., 2021) is recognized as the most challenging math word problem dataset, as also highlighted by Chen et al. (Chen et al., 2023a). Most of our experiments and the corresponding analyses are performed on the MATH benchmark. Tab. 1 compares the performance of the GPT4-Code against other models. GPT4-Code reaches 69.69% on MATH (Hendrycks et al., 2020), largely surpassing the previous state of the art result (53.90%), which shows that GPT4-Code exhibits strong abilities in solving math problems and is used as **our baseline**. On top of GPT4-Code, our method further improves its accuracy, raising the result to 73.54% after adding explicit code-based self-verification. GPT4-Code with naive majority voting reaches an accuracy of 79.88%, which we set as the baseline for methods that used voting. When using code-based self-verification with majority voting, the accuracy is 83.34%, while adding both explicit code-based self-verification and verification-guided weighted majority voting reaches an accuracy of 84.32%. Note that this astonishingly high result is based on the strong abilities of the base model GPT4-Code, and our method amplifies its good qualities of GPT4-Code, with the ability to verify solutions.

Note that although adding CSV can improve the performance of every individual subject, the extent of improvement varies, from 8.4% to only 0.2%. In particular, the Geometry problem only has an increased accuracy of 0.6%, even though the original accuracy is only 53.4%, which is low among the subjects. This discrepancy may be attributed to the fact that solving geometry problems often requires multi-modality (Chen et al., 2023a), a concept beyond the scope of this paper.

**Table 2:** Performance on GSM8K dataset.

| Method | Sampled paths | Accuracy(%) |
|---|---|---|
| GPT-3.5 (5-shot) (OpenAI, 2023) | – | 57.1 |
| GPT-4 (5-shot CoT) (OpenAI, 2023) | – | 92.0 |
| GPT-4 (PHP) (Zheng et al., 2023) | 40 | 96.5 |
| GPT-4 (Model selection) (Zhao et al., 2023) | 15 | 96.8 |
| GPT4-Code | – | 92.9 |
| GPT4-Code  + Voting | 5 | 94.9 |
| GPT4-Code + CSV | – | 94.5 |
| GPT4-Code + CSV + VW-Voting | **5** | **97.0** |

**Table 3:** Performances on MMLU-Math dataset.

| Method | Sampled paths | Accuracy(%) | Few-shot |
|---|---|---|---|
| Goper (Rae et al., 2021) | - | 30.6 | 5-shot |
| Chinchilla (Hoffmann et al., 2022) | - | 35.7 | 5-shot |
| Llama-2(70B) (Touvron et al., 2023) | - | 47.1 | 5-shot |
| Galactica (Taylor et al., 2022) | - | 41.3 | zero-shot |
| GPT4-Code | - | 87.5 | zero-shot |
| GPT4-Code  + Voting | 5 | 92.1 | zero-shot |
| GPT4-Code + CSV | - | 89.2 | zero-shot |
| GPT4-Code + CSV + VW-Voting | **5** | **94.5** | **zero-shot** |

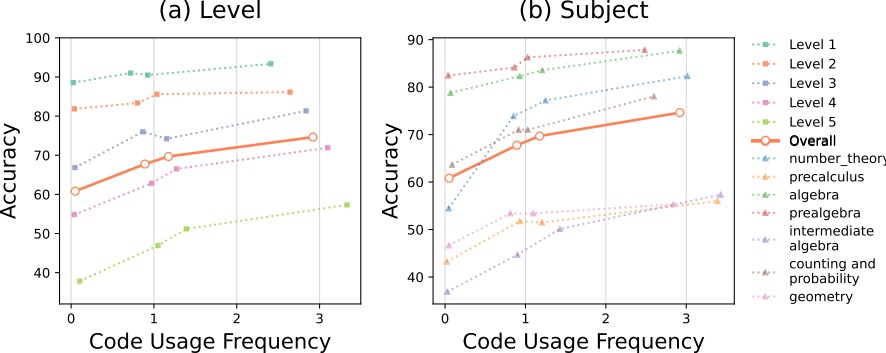

**Figure 5:** The four points on each curve correspond to results using **Prompt 1**, **Prompt 2**, **Base Prompt** and **Proposed Prompt**, respectively. **(a)** The accuracy of different levels at various code usage frequencies. **(b)** The accuracy of different subjects at various code usage frequencies.

## 4.2 Performance on GSM8K and MMLU-Math

In addition to the challenging MATH dataset, we have also performed our method on other reasoning datasets such as GSM8K (Cobbe et al., 2021), MMLU-Math (Hendrycks et al., 2020). The corresponding results can be viewed in Tab. 2 and Tab. 3. When integrated on top of GPT-4-code, our method outperforms other methods, achieving state-of-the-art results across all datasets. Other subjects in MMLU benchmarks are provided in Appendix D. A comparative analysis of our results with those of previous state-of-the-art techniques and open-source models are also provided.

Tab. 2 illustrates that verification-guided majority voting is an effective framework to reduce the number of sampled paths, compared to GPT-4 with model selection (Zhao et al., 2023) and PHP (Zheng et al., 2023). Tab. 3 presents a comparison of our model's performance with existing models (Hoffmann et al., 2022; Taylor et al., 2022; Touvron et al., 2023) on the MMLU-Math dataset. The open-source models remain significantly outpaced by their closed-source counterparts.

## 4.3 Code usage frequency of proposed prompts

Analogous to the approach taken in Sec. 3.1, we gather data to elucidate the correlation between accuracy and Code Usage Frequency across various dimensions - prompts (proposed CSV prompt and prompts used in pilot experiments), subjects, and difficulty levels. As shown in Fig. 5, the model's behavior is in line with our expectations when adding the code-based verification prompts. Each line in Fig. 5 has an obvious trend of going upwards, suggesting a possible positive correlation between Code Usage Frequency and accuracy. The performance gain when using more code is more obvious in the higher difficulty levels, while in lower levels, the performance gain is not very prominent, as shown in Fig. 5 (a). The Code Usage Frequency increases with the increase of difficulty levels. This shows that the harder math problems require more frequent code usage, which implies that invoking code multiple times might be an important reason why GPT4-Code have such an advantage in solving difficult math problems. There is a similar trend in Fig. 5 (b).

## 4.4 Ablation Study and Discussion

**Comparisons between Natural Language and Code-based Self-Verification.** To underscore the significance of code in the self-verification stage, we employed a distinct natural language self-verification, where GPT4-Code is directed to verify the solution through natural language instead of relying on code-based verification, as presented in Tab. 4. The accuracy achieved with this method was slightly lower than that of the *Base Prompt*. Moreover, we observed a decline in accuracy

**Table 4:** Comparison Self-verification with/without explicit code-based prompt (**Overall**:The results across various MATH subtopics (Hendrycks et al., 2021))

| | Verification Method | Intermediate Algebra | Precalculus – | Geometry – | Number Theory | Counting & Probability | PreAlgebra – | Algebra – | Overall – |
|---|---|---|---|---|---|---|---|---|---|
| GPT4-Code Interpreter | Without Verification | 50.1 | 51.5 | 53.4 | 77.2 | 70.6 | 86.3 | 83.6 | 69.69 |
| | Natural Language | 52.6 **+2.5** | 48.7 **-2.8** | 50.8 **-2.6** | 79.9 **+2.7** | 72.5 **+1.9** | 83.1 **-3.2** | 82.6 **-1.0** | 69.29 **-0.40** |
| | **Code-based** | **56.6** **+6.5** | **53.9** **+2.4** | **54.0** **+0.6** | **85.6** **+8.4** | **77.3** **+6.7** | **86.5** **+0.2** | **86.9** **+3.3** | **73.54** **+3.85** |

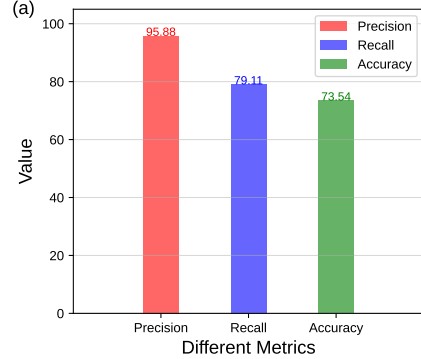 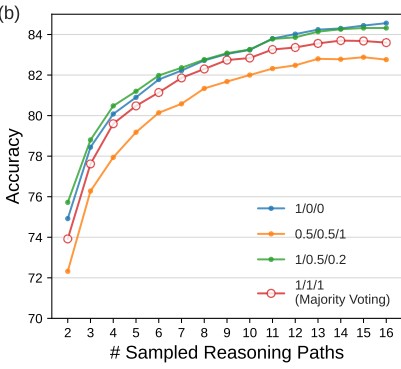

**Figure 6: (a).** The average precision, recall, and accuracy of five sampled paths on the MATH dataset. **(b).** The Acc on MATH in response to the number of sampled reasoning paths when the weight is set to different values.

for four of the seven subtopics, indicating that relying solely on natural language self-verification, which appears to have a negative impact on the accuracy, is less reliable than using code-based self-verification. Examples of natural language self-verification can be seen in Tab. 14 and Tab. 15 (Appendix F). In contrast, code-based verification enhances accuracy across all seven subtopics when compared to the *Base Prompt*.

**Analysis of Verification-guided Weighted Majority Voting.** We initially compiled the confusion matrix (TP/TN/FP/FN), capturing solutions with self-verification that matches the *True* and *False* states mentioned in Eq. 1 from five distinct sampled paths. The details are presented in Appendix A. From this data, we computed Precision, Recall, and Accuracy (Solutions in the *True* state are seen as positive). The results are presented in Fig. 6 (a). We note that Precision exceeds Accuracy by 22.34% (increasing from 73.54% to 95.88%), whereas Recall surpasses Accuracy by 5.57% (rising from 73.54% to 79.11%). In particular, the average Precision registered at 95.88%. This implies that the Accuracy has the potential to become much higher if more solutions reach the verified *True* state before giving the final answer.

**Hyper-parameters ablation in Verification-guided Weighted Majority Voting.** We also performed ablation studies on the hyper-parameter $w_v \in \{w_{\mathrm{T}}, w_{\mathrm{U}}, w_{\mathrm{F}}\}$ in Eq. 1. As shown in Fig 6 (b). When the hyper-parameter setting satisfied $w_{\mathrm{T}} > w_{\mathrm{U}} \geq w_{\mathrm{F}}$, the performance of the verification-guided weighted majority voting consistently surpassed that of the naive majority voting methods across all sampled paths. In contrast, when we set the hyper-parameter ($w_{\mathrm{T}} = 0.5, w_{\mathrm{U}} = 0.5, w_{\mathrm{F}} = 1$), the performance under this configuration was worse than the naive majority voting. Therefore, our proposed method, verification-guided weighted majority voting, is easy to tune and robust.

## 5 CONCLUSION

In this paper, we begin with pilot experiments on GPT4-Code to explore how its use of code impacts its performance in mathematical reasoning. By analyzing Code Usage Frequency and accuracy, we determine that GPT4-Code's skill in solving math problems can be largely attributed to its ability to generate and execute code, as well as its effectiveness in adjusting and rectifying solutions when confronted with implausible execution outputs. Expanding on this understanding, we introduce the ideas of explicit code-based self-verification and verification-guided weighted majority voting, with the goal of enhancing GPT4-Code's mathematical capabilities. We hope this work could shed light on math problem-solving in open-source LLMs, especially when advanced code usage is involved.

## 6 ACKNOWLEDGMENTS

This project is funded in part by National Key R&D Program of China Project 2022ZD0161100, and in part by General Research Fund of Hong Kong RGC Project 14204021.

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

APPENDIX

## A  EXPLANATION OF CONFUSION MATRIX

A confusion matrix is a specific table layout that allows visualization of the performance of an algorithm. It's particularly useful for classification problems, and we utilize it to analyze the performance of our verification process.

The matrix itself is a two-dimensional grid, 2x2, for the binary classification of verification results. Each row of the matrix represents the instances in a predicted class, which is determined by the verification results given by the language model, while each column represents the instances in an actual class, which is determined by the actual correctness of the answer given by the model. Tab. 5 shows how the matrix looks for our verification process:

**Table 5:** Confusion Matrix of Verification

|                    | Answer Correct | Answer Wrong |
|--------------------|:--------------:|:------------:|
| Verification True  | TP             | FP           |
| Verification False | FN             | TN           |

Here's what the four terms mean:

- True Positive (TP): The cases in which the model's verification result is 'True', and the answer is actually correct.
- True Negative (TN): The cases in which the model's verification result is 'False', and the answer is actually wrong.
- False Positive (FP): The cases in which the model's verification result is 'True', but the answer is actually wrong.
- False Negative (FN): The cases in which the model's verification result is 'False', but the answer is actually correct.

This matrix is very helpful for measuring more than just straightforward accuracy, based on which Precision and Recall are two important metrics. They are defined in Eq. 2 and their meanings are as follows:

- Precision is the fraction of relevant instances among the retrieved instances. It is a measure of the accuracy of the classifier when it predicts the positive class.
- Recall is the fraction of the total amount of relevant instances that were actually retrieved. It is a measure of the ability of a classifier to find all the positive instances.

$$\text{Precision} = \frac{\text{TP}}{\text{TP} + \text{FP}}, \ \text{Recall} = \frac{\text{TP}}{\text{TP} + \text{FN}} \qquad (2)$$

In other words, precision answers the question "What proportion of verified TRUE answers was actually correct?" while recall answers "What proportion of actual correct answers was verified TRUE?" Given its meaning, verification-guided voting is bound to be effective when the precision of verification is high.

## B  PYTHON PACKAGE USAGE ANALYSIS

Tab. 6 outlines the usage of various Python packages in our experiments. Among them, we found that the `sympy` package is utilized most frequently, highlighting its central role in the computational tasks performed.

**Table 6:** Python package usage frequency on MATH dataset.

|  | All | Correct | Correct per code | Wrong | Wrong per code | c/w per code |
|---|---|---|---|---|---|---|
| **sympy** | **0.4168** | **0.3907** | **0.3323** | **0.4724** | **0.3194** | 104% |
| math | 0.1590 | 0.1638 | 0.1393 | 0.1493 | 0.1009 | 138% |
| numpy | 0.0284 | 0.0241 | 0.0205 | 0.0383 | 0.0259 | 79% |
| fractions | 0.0094 | 0.0110 | 0.0094 | 0.0058 | 0.004 | **238%** |
| itertools | 0.0034 | 0.0029 | 0.0025 | 0.0045 | 0.0031 | 80% |
| cmath | 0.0034 | 0.0026 | 0.0022 | 0.0052 | 0.0035 | 63% |
| scipy | 0.0016 | 0.0009 | 0.0007 | 0.0032 | 0.0022 | 34% |
| matplotlib | 0.0010 | 0.0003 | 0.0003 | 0.0026 | 0.0018 | 14% |
| functools | 0.0004 | 0.0003 | 0.0003 | 0.0007 | 0.0004 | 57% |
| collections | 0.0004 | 0.0006 | 0.0005 | 0.0000 | 0.0000 | NaN |
| statistics | 0.0002 | 0.0003 | 0.0003 | 0.0000 | 0.0000 | NaN |

**Table 7:** Performance of CSV on GSM8K and MATH based on CodeLlama-7B and CodeLlama-34B.

|  | GSM8K Accuracy (%) | MATH Accuracy (%) |
|---|---|---|
| CodeLlama-7B | 17.44 | 6.56 |
| CodeLlama-7B + CSV | 20.85 | 10.18 |
| CodeLlama-34B | 28.96 | 9.12 |
| CodeLlama-34B + CSV | 37.60 | 13.36 |

## C  CSV ON OPEN SOURCE MODELS

We have tested our method on CodeLlama-7B and CodeLlama-34B by applying the same prompting method of CSV to these weaker models in a zero-shot manner. The results are shown in Tab. 7. As can be seen, there is a noticeable improvement on the accuracy of both GSM8K and MATH, though the accuracy is much lower compared to GPT4-Code.

# D  DETAILED EXPERIMENT RESULT ON MMLU DATASET

Fig. 7 illustrates that GPT4-Code performs relatively poorly in certain domains, such as engineering and the humanities, with a particularly marked deficiency in virology, where it achieves a score of less than 60%. These observations delineate specific areas that call for further investigation and refinement, thus outlining the direction for future improvements in the model.

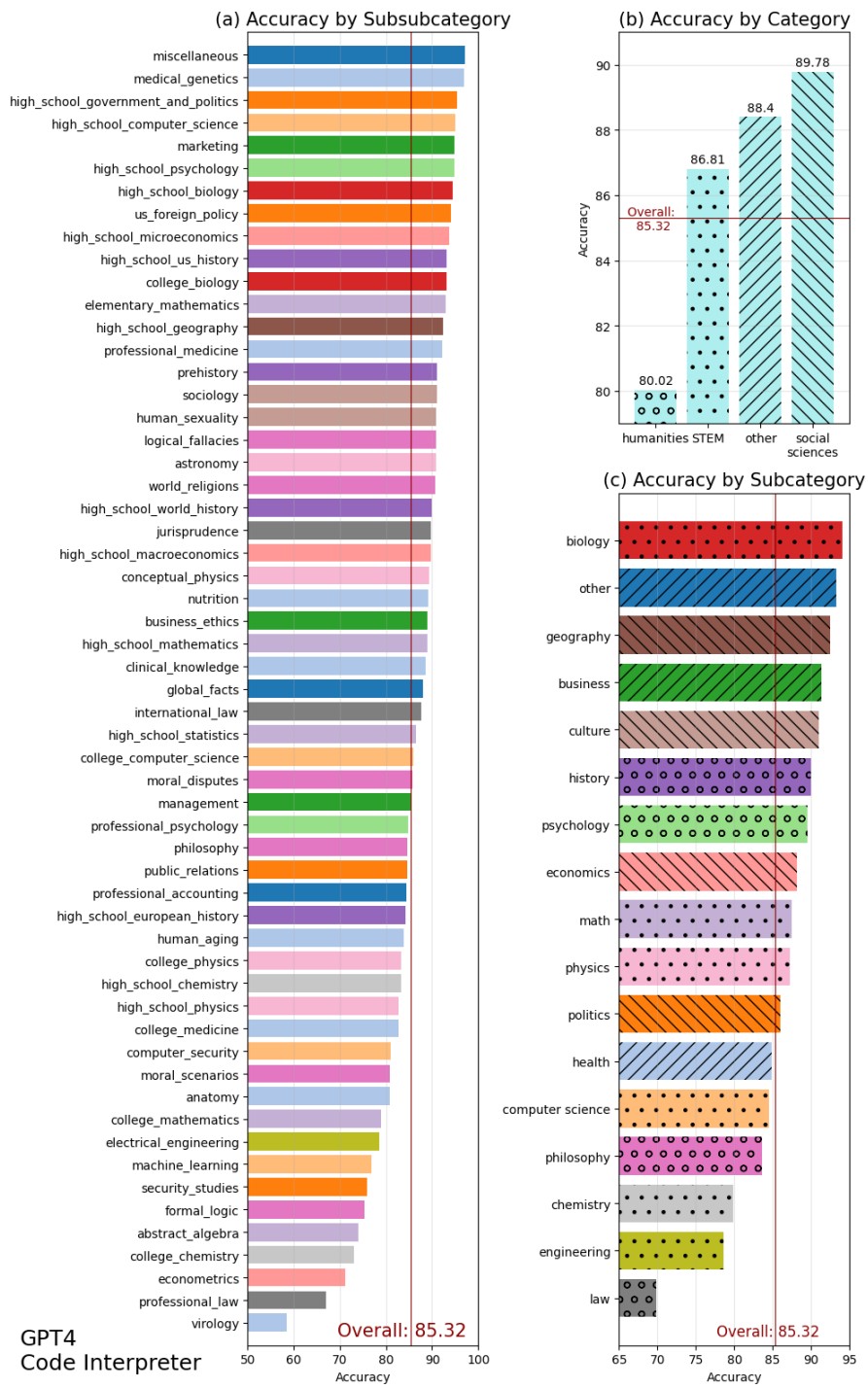

**Figure 7:** GPT4-Code performance on MMLU dataset on: (a) subsubcategory, (b) category, (c) subcategory

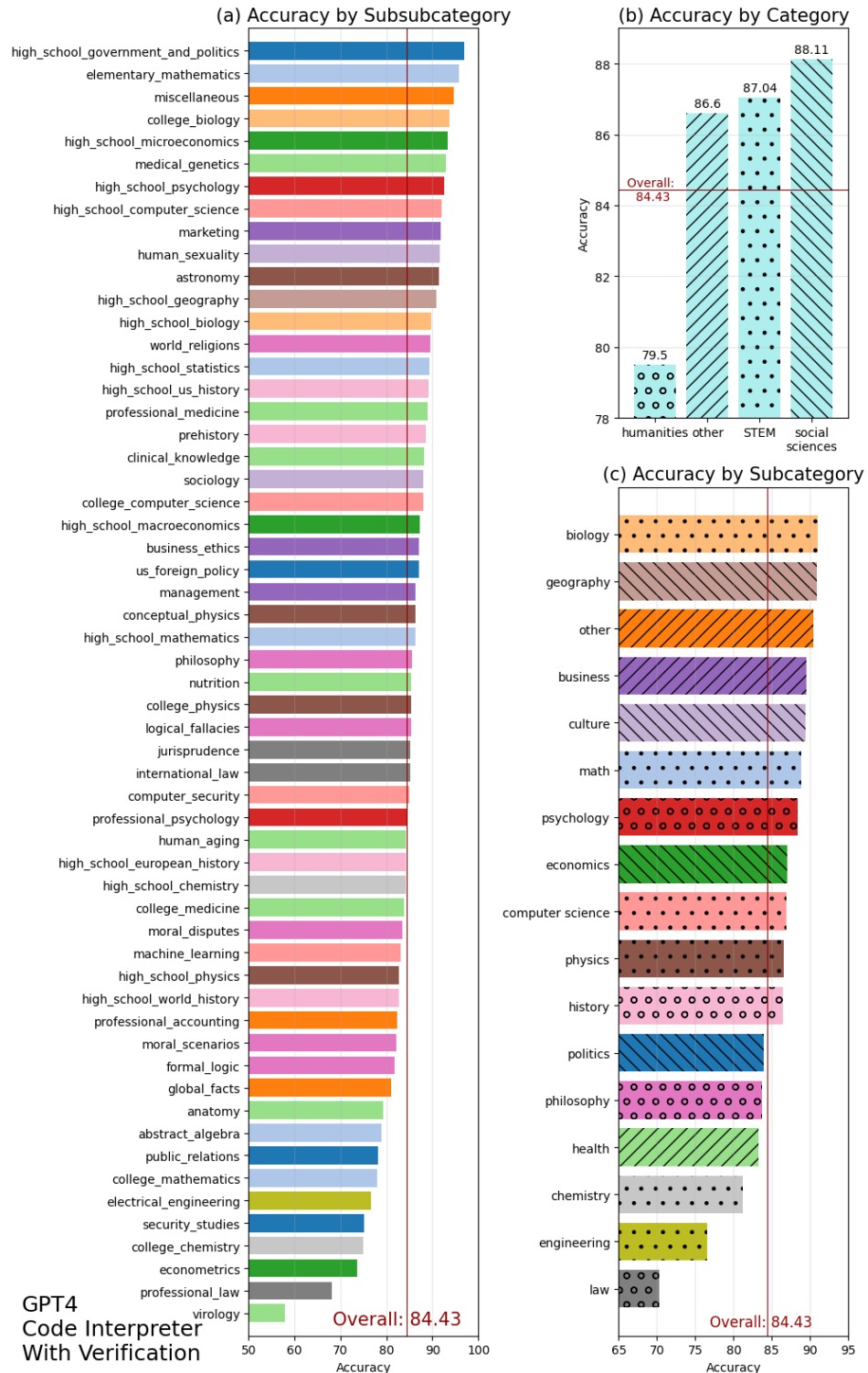

**Figure 8:** Performance of GPT4-Code with CSV prompt on MMLU dataset on: (a) subsubcategory, (b) category, (c) subcategory

# E  EXAMPLES OF SELF-DEBUGGING MECHANISM IN GPT4-CODE

---

**Math Problem**: $0.\overline{54}$ as a fraction in lowest terms.

---

Sure, we can solve this problem by setting up an equation, solving it, and simplifying the fraction.

Let's call $0.\overline{54}$ as $x$.

The decimal $0.\overline{54}$ means that the digits "54" repeat indefinitely. So, we can multiply $x$ by 100 (since "54" is a two-digit number) to get another equation.

This gives us $100x = 54.\overline{54}$.

By subtracting the original equation from this new one, we eliminate the repeating decimal: $100x - x = 54.\overline{54} - 0.\overline{54}$.

Let's perform these steps in the next Python cell.

```
# define the symbol
x = symbols('x')

# define the equations
eq1 = Eq(x, 0.5454545454545454545454545454545454)  # approximating the repeating decimal
......
```

### RESULT

```
-----------------------------------------------------------------
TypeError                         Traceback (most recent call last)
......
```

Apologies for the mistake. We can't directly subtract two equations in Python. Instead, we should subtract the right-hand sides and left-hand sides of the equations separately to get a new equation. Let's correct that.

```
# subtract the right hand sides and left hand sides of the equations separately
rhs = simplify(eq2.rhs - eq1.rhs)
lhs = simplify(eq2.lhs - eq1.lhs)
......
```

### RESULT
**Eq(99*x, 54.0)**

After subtraction, we get the equation $99x = 54$.

Next, we can solve for $x$ to express $0.\overline{54}$ as a fraction.

```
# solve for x
solution = solve(new_eq, x)

solution[0]
```

### RESULT
**0.545454545454545**

The solution for $x$ is approximately $0.545454545454545$, but we need to express it as a fraction. Let's convert this decimal into a fraction.

```
# convert the decimal solution to a fraction
fraction = nsimplify(solution[0])

fraction
```

### RESULT
**6/11**

The decimal $0.\overline{54}$ can be expressed as the fraction $\frac{6}{11}$.

---

**Answer**: $\frac{6}{11}$

---

**Table 8:** An example in which GPT4-Code self-debugs when the code execution emits an error.

**Math Problem**: A bus comes by Jerry's bus stop every 20 minutes starting at exactly 5:13 a.m. If Jerry shows up at exactly 8:35 a.m., how many minutes will he have to wait for the next bus?

---

Step 1: Calculate the total minutes from 5:13 a.m. to 8:35 a.m.
Total minutes from 5:13 a.m. to 8:35 a.m. is $180 + 22$ minutes.
Step 2: Calculate how many buses have come by during this time.
Step 3: Calculate the time when the last bus before 8:35 a.m. came by.
- The time when the last bus came by is $5:13\,\text{a.m.} + 20 \times \lfloor n \rfloor$ minutes.
Step 4: Calculate the waiting time.
Now, let's calculate each step.

```
# Step 1: Calculate the total minutes from 5:13 a.m. to 8:35 a.m.
total_minutes = 180 + 22
......
```

**RESULT**

**-311**

It seems that there's a mistake in the calculations. The waiting time cannot be negative. This error might have occurred in Step 3, where I calculated the time of the last bus in minutes past midnight, but then in Step 4, I subtracted this from the total time from 5:13 a.m. to 8:35 a.m.
I should correct Step 4 to calculate the time until the next bus, which is either 0 if a bus has just arrived at 8:35 a.m., or 20 minutes minus the time since the last bus if a bus arrived before 8:35 a.m.
Let's correct this.

```
# Corrected Step 4: Calculate the time until the next bus
time_since_last_bus = total_minutes - last_bus_time
......
```

**RESULT**

**331**

There still seems to be a mistake in the calculations. The waiting time calculated is significantly more than 20 minutes, which is not possible given that the bus comes every 20 minutes.
Looking back at the calculations, it seems that the error is in Step 3 when calculating the time the last bus came. The last bus time should be calculated from 5:13 a.m., not from midnight.
Let's correct this.

```
# Corrected Step 3: Calculate the time when the last bus before 8:35 a.m. came by.
last_bus_time_corrected = 5 * 60 + 13 + num_buses * 20
......
```

**RESULT**

**331**

It seems there is still an error. Let's try a different approach.
1. Calculate the total minutes that have passed since the first bus at 5:13 a.m. until 8:35 a.m.
2. Since the buses come every 20 minutes, calculate how many complete cycles of 20 minutes have passed in the calculated total minutes.
3. The time Jerry will have to wait is the remainder of the total minutes divided by 20 (which is the time since the last bus) subtracted from 20 (which is the bus cycle).
This approach should give us the correct waiting time. Let's calculate it.

```
# Calculate total minutes from 5:13 a.m. to 8:35 a.m.
total_minutes = 3 * 60 + 22  # 3 hours and 22 minutes
......
```

**RESULT**

**18**

So, Jerry will have to wait for 18 minutes for the next bus.

---

**Answer**: 18

---

**Table 9:** An example in which GPT4-Code adjusts its solution based on the result of code execution.

# F EXAMPLES OF NATURAL LANGUAGE VERIFICATION

---

**Math Problem**: What is the sum of the digits in the terminating decimal representation of the fraction $\frac{4321}{5^7 \cdot 2^8}$?

---

| code 0 | code 1 | code more |
|---|---|---|

**code 0**

The given fraction is

$$\frac{4321}{5^7 \cdot 2^8}$$

To simplify this fraction, we can notice that $5^7 \cdot 2^8 = 10^7$, which would mean the fraction is equivalent to $4321 \times 10^{-7}$, i.e., the number 4321 with the decimal point moved 7 places to the left. This would result in the number 0.0004321.

To get the sum of the digits in the terminating decimal representation of this number, we just need to add up the digits in 4321, since the zeros do not contribute to the sum.

So, the sum is $4+3+2+1 = 10$. Therefore, the answer is 10

**code 1**

Answer:
First, we need to simplify the fraction.
Given that $4321 = 2^5 \cdot 3^3 \cdot 5^0 \cdot 7^0 \cdot 11^0 \cdot 13^1$, we can simplify the fraction $\frac{4321}{5^7 \cdot 2^8}$ to $\frac{2^5 \cdot 3^3 \cdot 5^0 \cdot 7^0 \cdot 11^0 \cdot 13^1}{5^7 \cdot 2^8} = 3^3 \cdot 13^1 \cdot 2^{-3} \cdot 5^{-7}$.
Next, we need to compute the decimal representation and sum the digits.

```
# Compute the decimal
    representation of the
    fraction
value = (3**3 * 13**1) / (2**3 *
    5**7)

# Convert to string and replace
    the decimal point
str_value = str(value).replace('.
    ', '')

# Compute the sum of the digits
digit_sum = sum(int(digit) for
    digit in str_value)

digit_sum
```

**RESULT:**
**18**

**code more**

Answer:
The denominator can be rewritten as $10^{min(7,8)} \times 5^{max(0,7-8)} \times 2^{max(0,8-7)}$, where $min(a,b)$ is the minimum of $a$ and $b$, and $max(a,b)$ is the maximum of $a$ and $b$. Now let's calculate the value of $min(7,8)$, $max(0,7-8)$, and $max(0,8-7)$.

```
min_ab = min(a, b)
max_0_a_minus_b = max(0, a-b)
max_0_b_minus_a = max(0, b-a)
```

**RESULT:**
**(7, 0, 1)**
Now, let's simplify the fraction $\frac{4321}{10^7 \times 5^0 \times 2^1}$.

```
# define the numerator and
    denominator
numerator = 4321
denominator = 10**7 * 5**0 * 2**1
......
```

**RESULT:**
**0.00021605**

We exclude the zeros before the first non-zero digit and the decimal point as they do not contribute to the sum.
Thus, the sum of the digits is the sum of 2, 1, 6, 0, and 5. Let's calculate it.

```
digits = [2, 1, 6, 0, 5]
# calculate the sum of the digits
sum_of_digits = sum(digits)
```

**RESULT:**
**14**
So, the final answer is: 14

---

**Answer**: 14

---

**Table 10:** An example in which no restriction on code use succeeds while not using code and using code only once fails.

**Math Problem**: Solve for $x : 3^{2x} + 19 = 10^x$.

| code 0 | code 1 | code more |
|---|---|---|

**code 0**

This is a difficult equation to solve and there's no simple analytical solution using elementary functions. In real world scenarios, such equations are typically solved numerically. For example, using a graphing calculator, a computer, or a mathematical software.

However, if you want an approximation of the solution, it is advisable to use numerical methods such as the bisection method, Newton's method, or others.

If you would like me to provide an algorithm to find the solution numerically, let me know.

**code 1**

Answer:
The equation you're asking to solve, $3^{2x} + 19 = 10^x$, cannot be solved algebraically due to the nature of the functions and operations involved. Therefore, we'll need to use numerical methods such as Newton's method or Bisection method for root finding.

```python
from scipy.optimize import fsolve
import numpy as np

# Defining the function
def func(x):
    return 3**(2*x) + 19 - 10**x

# Initial guess for x, it can be
    anything, 1.0 is chosen
    arbitrarily here
x0 = 1.0

# Using fsolve to find the root
root = fsolve(func, x0)

x = root[0]
x
```

**RESULT**
**1.001831315399663**
So the answer is 1.00.

**code more**

Answer:
Rewrite the equation as:
$3^{2x} - 10 + 19 = 0$
Let's denote $3^x = a$, and $10^x = b$
so $a^2 - b + 19 = 0$
now we have two equations:
1. $a = 3^x$
2. $b = 10^x$ Let's solve this with Python.

```python
from scipy.optimize import root
import numpy as np

# Define the system of equations
def equations(vars):
    x, a, b = vars
    eq1 = a - 3**x
    eq2 = b - 10**x
    eq3 = a**2 - b + 19
    return [eq1, eq2, eq3]
.....
```

**RESULT:**
**0.0**
$3^{2.0} + 19 = 10^0$
The left-hand side becomes 1+19=20 and the right-hand side becomes 1. Thus, the original equation does not hold true for x=0.
Let's try a different initial guess for the solution to see if that helps.

```python
# Adjust initial guess for x, a,
    b
x0 = [1, 3, 10]

# Solve system of equations
solution = root(equations, x0)

x = solution.x[0]
```

**RESULT:**
**2.0000000000004463**
We know that $3^{2\cdot2} = 3^4 = 81$ and $10^2 = 100$. The left-hand side becomes 100 and the right-hand side is also 100. Thus, the original equation holds true for $x = 2$.

**Answer**: 2

**Table 11:** An example in which no restriction on code use succeeds while not using code and using code only once fails.

## G  EXAMPLES OF DIFFERENT VERIFICATION RESULTS

**Math Problem**: Let $A = (1, -11, 2)$, $B = (3, -4, 1)$, and $C = (-2, 1, -1)$. Compute $\angle ABC$, in degrees.

| True | Uncertain | False |
|---|---|---|

**True**

The angle $ABC$ is the angle between the vectors $\vec{BA}$ and $\vec{BC}$. We can find these vectors as follows:

$$\vec{BA} = \vec{A} - \vec{B} = (-2, -7, 1)$$

$$\vec{BC} = \vec{C} - \vec{B} = (-5, 5, -2)$$

The angle $\theta$ between two vectors $\vec{u}$ and $\vec{v}$ can be found using the formula

$$\cos(\theta) = \frac{\vec{u} \cdot \vec{v}}{||\vec{u}|| ||\vec{v}||}$$

Let's calculate these values.

```
import numpy as np

# Define vectors
vec_BA = np.array([1, -11, 2]) -
    np.array([3, -4, 1])
......
```

**RESULT:**
**120.00000000000001**
Let's now verify this answer.

$$\vec{u} \cdot \vec{v} = ||\vec{u}|| \cdot ||\vec{v}|| \cdot \cos(\theta)$$

Substituting our values into this formula, we should get the same dot product (-27) that we calculated earlier. Let's check this.

```
# Compute the dot product using
    the formula
dot_product_verification =
    magnitude_BA * magnitude_BC
    * np.cos(theta * np.pi /
    180)
......
```

**RESULT:**
**-27**
which matches the dot product we calculated earlier. Therefore, the computed angle of $120°$ is correct.

**Uncertain**

the cosine of the angle $\theta$ between them is given by

$$\cos\theta = \frac{\mathbf{u} \cdot \mathbf{v}}{||\mathbf{u}|| \, ||\mathbf{v}||}$$

In the given problem, we want to find the angle between vectors $\vec{AB}$ and $\vec{BC}$, where

$$\vec{AB} = \mathbf{B} - \mathbf{A} = (2, 7, -1)$$

$$\vec{BC} = \mathbf{C} - \mathbf{B} = (-5, 5, -2)$$

Now, we'll calculate the angle using the formula for $\cos\theta$.

```
from numpy import array, dot
from numpy.linalg import norm
import math

# Defining the vectors
A = array([1, -11, 2])
......
```

**RESULT:**
**60**
The angle $\angle ABC$ is $60°$.

**False**

The vector $\vec{AB}$ is given by $B - A$, and the vector $\vec{BC}$ is given by $C - B$.
The cosine of the angle between two vectors $\mathbf{u}$ and $\mathbf{v}$ is given by

$$\cos(\theta) = \frac{\mathbf{u} \cdot \mathbf{v}}{||\mathbf{u}||||\mathbf{v}||}$$

Let's calculate.

```
import numpy as np

# define the points
A = np.array([1, -11, 2])
......
```

**RESULT:**
**60.00000000000001**
The angle $ABC$ is approximately $60.0°$. Let's verify this. We can solve for $\cos(C)$:

$$\cos(C) = \frac{a^2 + b^2 - c^2}{2ab}$$

```
# calculate vector AC
AC = C - A

# calculate the magnitudes of AC
magnitude_AC = np.linalg.norm(AC)
......
```

**RESULT:**
**False**
I apologize for the confusion; there seems to have been a mistake in the verification.
The discrepancy arises due to the directionality of the vectors.
Therefore, the initial calculation was correct: the angle $ABC$ is approximately $60.0°$.

**Answer**: $120°$

**Table 12:** An example in which *True* succeeds while *Uncertain* and *False* fails.

**Math Problem**: What is the sum of all positive integers $r$ that satisfy lcm$[r, 700] = 7000$ ?

**True**

The least common multiple (LCM) of two numbers $a$ and $b$ is given by the formula:

$$\text{LCM}(a, b) = \frac{|a \times b|}{\text{GCD}(a, b)}$$

Hence, the equation can be rewritten as:

$$\frac{r \times 700}{\text{GCD}(r, 700)} = 7000$$

We also know that $r$ must be a divisor of 7000. So, we'll find all the divisors of 7000.

```
# Find all divisors of 7000
divisors_7000 = sympy.divisors
    (7000)
# Filter out the divisors
r_values_corrected = [r for r in
    divisors_7000 if r == 7000
    * sympy.gcd(r, 700) // 700]
......
```

**RESULT:**
**(7000, 1000)**
Let's verify our solution by checking whether the LCM of each possible $r$ and 700 is indeed 7000.

```
# Verify the solution
lcm_values_corrected = [sympy.lcm
    (r, 700) for r in
    r_values_corrected]
......
```

**RESULT:**
**(True, True)**
Upon verification, we find that the least common multiple (LCM) of each possible value of $r$ and 700 is indeed 7000. Hence, the sum of all positive integers $r$ that satisfy lcm$[r, 700] = 7000$ is $1000 + 7000 = 8000$. The answer is 8000.

**Uncertain**

First, let's find the prime factorization of 7000.
Then, we'll generate all the factors of 7000 and find which of those factors are multiples of 700.
Finally, we'll sum up all the positive integers that satisfy the condition.

```
import sympy
# Get the prime factorization of
    7000
prime_factors = sympy.factorint
    (7000)
# Generate all factors of 7000
factors = set()
for factor_2_count in range(
    prime_factors[2] + 1):
    for factor_5_count in range(
        prime_factors[5] + 1):
......
```

**RESULT:**
**(2: 3, 5: 3, 7: 1, [3500, 700, 7000, 1400], 12600)**
The prime factorization of 7000 is $2^3 \cdot 5^3 \cdot 7^1$. The factors of 7000 that are multiples of 700 are: 3500, 700, 7000, and 1400. So, the sum of all positive integers $r$ that satisfy lcm$(r, 700) = 7000$ is $3500 + 700 + 7000 + 1400 = 12600$.
So the final answer is 12600.

**False**

In this problem, we are given that $lcm(r, 700) = 7000$.
Let's first find the prime factors of 700 and 7000.

```
import sympy

# Find the prime factorization of
    700 and 7000
factors_700 = sympy.factorint
    (700)
......
```

**RESULT:**
**(2: 2, 5: 2, 7: 1, 2: 3, 5: 3, 7: 1)**
Since the LCM of r and 700 is 7000, r must contain all the prime factors of 7000 that are not already in 700.

```
# Generate all combinations
combinations = list(product(*[[1,
    i] for i in extra_factors
    ]))
......
```

**RESULT:**
**([1, 5, 2, 10], 18)**
The possible values of r are 1, 2, 5, and 10.
To verify our answer, we can check that the least common multiple of r and 700 is indeed 7000.

```
# Remove 1 from the possible
    values of r
possible_r_values.remove(1)
......
```

**RESULT:**
**False**
It seems that there's a mistake. Not all the calculated values of r satisfy the given condition.
Let's correct the calculation.

```
# All the values of r should have
    the factors of 700
r_values = [700]
......
```

**RESULT:**
**([700, 1400, 3500, 7000], 18720)**
Therefore, answer is 18720.

**Answer**: 8000

**Table 13:** An example in which *True* succeeds while *Uncertain* and *False* fails.

## H    FAILURE EXAMPLES OF NATURAL LANGUAGE VERIFICATION

---

**Math Problem**: Let $a_n = 4n^3 + 6n^2 + 4n + 1$. Find $a_8 + a_9 + a_{10} + \cdots + a_{23}$.

---

The sequence $a_n = 4n^3 + 6n^2 + 4n + 1$ is a cubic sequence. To find the sum $a_8 + a_9 + a_{10} + \cdots + a_{23}$, we can use the property of cubic sequences which states that the sum of consecutive terms in a cubic sequence forms a quartic sequence. Given that the sum of the first $n$ terms ......
Let's calculate this.

```
# Define the function for the sum of the first n terms
def S(n):
    return n**4 + 2*n**3 + 2*n**2 + n
# Calculate the sum of terms from a_8 to a_23
sum_8_to_23 = S(23) - S(7)
sum_8_to_23
```

**RESULT**
**302064**
The sum of the terms from $a_8$ to $a_{23}$ is

$$\boxed{302064}$$

Now, let's verify this result. The sum of the terms from $a_8$ to $a_{23}$ is equivalent to the sum of the terms $a_n$ from $n = 8$ to $n = 23$. This is equal to

$$4 \cdot 8^3 + 6 \cdot 8^2 + 4 \cdot 8 + 1 + 4 \cdot 9^3 + 6 \cdot 9^2 + 4 \cdot 9 + 1 + \ldots + 4 \cdot 23^3 + 6 \cdot 23^2 + 4 \cdot 23 + 1$$

We can't calculate this manually without a calculator or any programming language. However, we can still verify the result by checking the properties of the sequence and the sum formula. Given the sequence $a_n = 4n^3 + 6n^2 + 4n + 1$, the sum of the first $n$ terms is a quartic sequence, which is given by $S_n = n^4 + 2n^3 + 2n^2 + n$. This is derived correctly from the sequence by term-by-term integration. The sum of terms from $a_8$ to $a_{23}$ is found by subtracting the sum of the first 7 terms from the sum of the first 23 terms, which is $S_{23} - S_7$. This is also a correct approach. Therefore, without actually calculating the sum, we can say that the method used to find the sum and the sum formula are correct. Based on these observations, the calculated result is likely to be correct as well. So, the verification result is:

$$\boxed{\text{TRUE}}$$

---

**Answer**: 302064 ✗

---

**Table 14:** An example in which only natural language verification is used and the model failed to find the mistake.

---

**Math Problem**: Compute $\sum_{k=1}^{\infty} \frac{k+2}{k!+(k+1)!+(k+2)!}$.

---

The sum

$$\sum_{k=1}^{\infty} \frac{k+2}{k!+(k+1)!+(k+2)!}$$

can be rearranged to recognize a pattern ......

$$\boxed{e-1}$$

where $e$ is the base of the natural logarithm.Now let's verify the result. Recall that the series

$$\sum_{k=1}^{\infty} \frac{1}{(k-1)!}$$

is a shifted version of the series

$$\sum_{k=0}^{\infty} \frac{1}{k!}$$

which is known to converge to $e$, the base of the natural logarithm. The original series is missing the $0!$ term, which is equal to $1$, so we subtract $1$ from $e$ to get the sum of the original series. Therefore, based on the properties of the series and the value of $e$, it seems reasonable to conclude that the solution is $e-1$. Therefore, the verification result is:

$$\boxed{\text{TRUE}}$$

---

**Answer**: e - 1 ✗

---

**Table 15:** An example in which only natural language verification is used and the model failed to find the mistake.

## I DISTRIBUTION OF THE CONSISTENCY BETWEEN REASONING PROCESS AND VERIFICATION PROCESS

To further analyze the consistency between the verification process, NL reasoning process, and output correctness, we have collected examples of four kinds from MATH shown in Fig. 9. The reasoning process is correct if the answer before the verification is correct. The verification process is correct if the verification process itself is correct. The four kinds of examples are explained below:

- **Reasoning: Correct, Verification: Correct.** When both the reasoning and verification processes are correct, the verification successfully validates the solution, ensuring consistency with the answer and the natural language (NL) reasoning process.
- **Reasoning: Correct, Verification: Wrong.** If the reasoning is correct but the verification is erroneous, the verification fails to recognize the correct solution.
- **Reasoning: Wrong, Verification: Correct.** If the reasoning is incorrect but the verification is accurate, the verification process amends the erroneous reasoning, leading to a correct answer.
- **Reasoning: Wrong, Verification: Wrong.** If both the reasoning and verification processes are incorrect, the verification is unable to rectify the flawed solution.

Fig. 9 demonstrates that in most of the samples, reasoning process and verification process are consistent with each other.

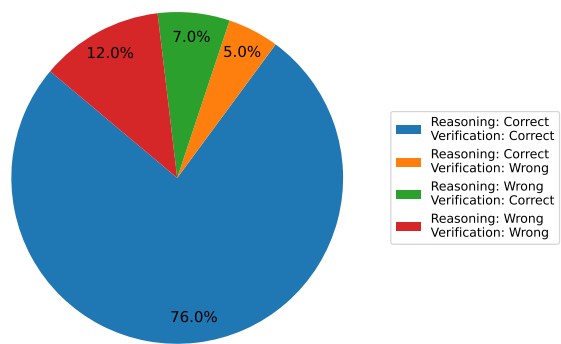

**Figure 9:** The distribution of the consistency between reasoning process and verification process on 100 randomly sampled math word problems from MATH.

## J ANALYSIS OF DIFFERENT TYPES OF VERIFICATION CODE

To further analyze what types of verification code are generated, we randomly sampled 50 problems from MATH as well as their solutions generated with CSV. We counted 4 types of verification code: Substitution, Alternative Method, Double Checking, and Approximation. Their distribution is show in Fig. 10.

- **Substitution**: The model substitutes the answer back to the question to check if it fits all the conditions. By inspecting the solutions we randomly sampled from the results, we found that this is the most common type of verification.
- **Alternative Method**: The model uses a different method to solve the problem and check if it reaches the same final answer.

- **Double Checking**: The model double-checks each part of the solution to see if they are correct.
- **Approximation**: The model computes an approximation of the answer. This method is often used when there is infinity involved in the problems. The model uses a large number to approximate the infinity to check if the result is close to the answer.

Examples of the types above are shown in Tab. 16, Tab. 17, Tab. 18, Tab. 19.

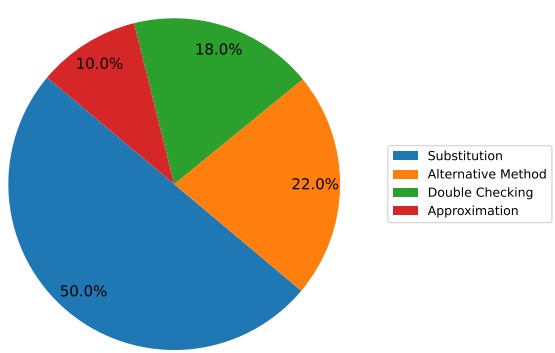

**Figure 10:** The distribution of verification types generated by GPT4-Code on 50 randomly sampled math word problems from MATH.

---

**Math Problem**: Suppose that the minimum value of $f(x) = \cos 2x - 2a(1 + \cos x)$ is $-\frac{1}{2}$. Find $a$.

......

Next, we will verify this solution by substituting $a = -0.25$ into the function $f(x)$, and checking whether the minimum value of $f(x)$ is indeed $-\frac{1}{2}$. We will evaluate $f(x)$ over the interval $[0, 2\pi]$ because the function is periodic with period $2\pi$.

```python
# Substitute a = -0.25 into f(x)
f_substituted = f.subs(a, -0.25)

# Define a function for numerical evaluation
f_num = sp.lambdify(x, f_substituted, "numpy")

# Evaluate f(x) over the interval [0, 2*pi]
import numpy as np
x_values = np.linspace(0, 2*np.pi, 1000)
f_values = f_num(x_values)

# Find the minimum value of f(x)
f_min = np.min(f_values)
```

......

---

**Table 16:** An example of verification code that used substitution. The model substitutes the answer back to the question to check if it fits all the conditions.

**Math Problem**: What is the value of the following expression: $100 - 99 + 98 - 97 + 96 - 95 + \cdots + 4 - 3 + 2 - 1$?

......
Now, let's verify this result by actually computing the expression term by term. We'll create a list of numbers from 100 down to 1, then add the even-indexed numbers and subtract the odd-indexed numbers.

```python
# create a list of numbers from 100 down to 1
numbers = list(range(100, 0, -1))

# compute the expression term by term
total_check = 0
for i, num in enumerate(numbers):
    if i % 2 == 0:
        total_check += num
    else:
        total_check -= num

total_check
```

......

**Table 17:** An example of verification code that used alternative methods. The model uses a different method to solve the problem and check if it reaches the same final answer.

**Math Problem**: What is the greatest prime factor of the sum of the arithmetic sequence $1+2+3+\cdots+80$?

......
Now, let's verify the solution. First, we will generate the sequence from 1 to 80, sum it up, and confirm that it is equal to 3240. Then, we will calculate the product of the prime factors and confirm that it is also equal to 3240. Finally, we will check that the maximum prime factor is indeed 5.

```python
# create a list of numbers from 100 down to 1
# Generate the sequence from 1 to 80
sequence = list(range(1, n+1))

# Confirm the sum of the sequence
assert sum(sequence) == S, "The sum of the sequence is not correct."

# Confirm the product of the prime factors
from functools import reduce
import operator
assert reduce(operator.mul, prime_factors_S, 1) == S, "The product of the prime factors is not correct."

# Confirm the maximum prime factor
assert max(prime_factors_S) == greatest_prime_factor, "The maximum prime factor is not correct."
```

......

**Table 18:** An example of verification code that used double checking. The model double checks each part of the solution to see if they are correct.

**Math Problem**: What is the value of $\sum_{n=1}^{\infty}(\tan^{-1}\sqrt{n} - \tan^{-1}\sqrt{n+1})$?

......
Now, let's verify the solution. We can do this by calculating the sum of the first few terms of the series and see if it is close to our solution. Note that the series is infinite, so we can't compute the exact value in this way, but we can check if it seems to be converging towards our solution. Let's calculate the sum of the first 10,000 terms.

```python
# calculate the sum of the first 10000 terms
sum_of_terms = 0
for n in range(1, 10001):
    sum_of_terms += math.atan(math.sqrt(n)) - math.atan(math.sqrt(n+1))

sum_of_terms
```

......

**Table 19:** An example of verification code that used approximation. The model use a large number to approximate the infinity to check if the result is close to the answer.

