# OpenReview forum: "Solving Challenging Math Word Problems Using GPT-4 Code Interpreter with Code-based Self-Verification"
_ICLR.cc/2024/Conference — ICLR 2024 poster_

### Official Review · Reviewer_UL1M · 2023-11-01

**Soundness:** 2 fair
**Presentation:** 4 excellent
**Contribution:** 1 poor
**Rating:** 3
**Confidence:** 5

**Summary:**

The paper attempts OpenAI's new GPT4 with Python Interpreter version. They reveal that incorporating code interpreter as an external could increase LLM's performance on math tasks. The more times using the interpreter, the higher performance would have. In addition, by using the self-debug method with the new GPT4 w/ code, they further push the SoTA of MATH dataset to 84.3%.

**Strengths:**

1. A good demonstration of OpenAI's new GPT4 with Python Interpreter version. The experiments show the great improvement of the new model comparing w/ the traditional GPT4. It also shows the potential of using code interpreter as an external tool to enhance performance on math reasoning tasks.
2. They authors push the SoTA of MATH to 84.3%, which is a very high number considering its complexity.

**Weaknesses:**

1. Most of the methods discussed in the paper are proposed by existing works and the main contribution of this paper is to try them out using the new OpenAI model. The contribution and novelty could be a weakness of the paper.
2. Most credit of the huge improvement on MATH should be given to the better capability of GPT4-code itself. The authors seem to over claim their own contribution throughout the paper. For example in Page 6, the authors said "Before the advent of GPT4-Code, prior frameworks (Lightman et al., 2023; Cobbe et al., 2021) depended on an external LLM to use natural language for verification and well-designed fewshot example prompts. In contrast, our approach simplifies the process by relying solely on a straightforward prompt for GPT4-Code, all in a **zero-shot** manner." But the root reason that zero-shot is applicable is because you are using OpenAI's instruction-tuned model. In addition, it is very likely that OpenAI has already used PRM (Lightman et al., 2023) to RL tune it's new model, therefore the API you used could be based on top of Lightman et al., 2023.

**Questions:**

N/A

---

> ### Author Response · Authors · 2023-11-20
> **Response to Official Review by Reviewer UL1M (1/3)**
>
> Thank you for your comments. To relieve your concerns, we have posted our responses below:
>
> **Q1**: Most of the methods discussed in the paper are proposed by existing works and the main contribution of this paper is to try them out using the new OpenAI model.
>
> **A1**: Thanks for your comments. Our method draws inspiration from existing works. However, we think that there is novelty in our discussion, analysis, and method, which we hope you could consider. We will discuss this in four aspects:
>
>
> First of all, we are the first to discover the important role code plays in GPT-4 Code Interpreter’s high performance in solving math problems via prompt engineering. GPT-4 Code Interpreter uses solutions interleaved with natural language, code, and execution results to solve math problems, possessing the ability to analyze execution results and self-repair. To the best of our knowledge, this is the first quantitative experimental analysis of this interesting phenomenon.
>
> Secondly, our method employs zero-shot prompting for automatic code-based self-verification, exploring GPT-4 Code Interpreter’s ability to use code multiple times, interleaved with natural language reasoning. Existing work on self-validation uses additional validator mechanisms ([3], [4], [5]), while our method employs zero-shot prompting for automatic self-verification. Additionally, our work uses code to verify the solution of math problems, which is different from existing work as well.
>
> Thirdly, we employ weighted voting based on the results from the proposed code-based self-validation, thereby improving the performance of existing majority voting, as demonstrated in Figure 6. In contrast, most existing works use majority voting. This verification-guided weighted majority voting has never been studied before.
>
> Fourthly, our paper focuses on the potential of using code multiple times, **interleaved with natural language reasoning**, while current methods using code, such as PAL [1] and PoT [2], generally only use code once.
>
> In summary,  although there are existing language models on code writing, self-validating, and majority voting, our method is notably different from theirs. We believe our approach of quantitative analysis and enhancing GPT4-Code with self-verification provides valuable and novel insights into the current top Large Language Models (LLMs).
>
> [1] Luyu Gao, Aman Madaan, Shuyan Zhou, Uri Alon, Pengfei Liu, Yiming Yang, Jamie Callan, and Graham Neubig. PAL: Program-aided language models. In International Conference on Machine Learning, pp. 10764–10799. PMLR, 2023.
>
> [2] Wenhu Chen, Xueguang Ma, Xinyi Wang, and William W. Cohen. Program of thoughts prompting: Disentangling computation from reasoning for numerical reasoning tasks, 2022.
>
> [3] Karl Cobbe, Vineet Kosaraju, Mohammad Bavarian, Mark Chen, Heewoo Jun, Lukasz Kaiser, Matthias Plappert, Jerry Tworek, Jacob Hilton, Reiichiro Nakano, Christopher Hesse, and John Schulman. Training verifiers to solve math word problems. arXiv preprint arXiv:2110.14168, 2021.
>
> [4] Hunter Lightman, Vineet Kosaraju, Yura Burda, Harri Edwards, Bowen Baker, Teddy Lee, Jan Leike, John Schulman, Ilya Sutskever, and Karl Cobbe. Let’s verify step by step. arXiv preprint arXiv:2305.20050, 2023.
>
> [5] Yixuan Weng, Minjun Zhu, Fei Xia, Bin Li, Shizhu He, Kang Liu, and Jun Zhao. Large language models are better reasoners with self-verification, 2023.
>
>
> **Q2**: Most credit of the huge improvement on MATH should be given to the better capability of GPT4-code itself. The authors seem to over claim their own contribution throughout the paper.
>
> **A2**: We are sorry that our paper has given you this impression. It is true that not all improvements are our contributions, which we have pointed out in our paper. We pointed out that GPT4-Code itself can achieve remarkable accuracy (69.69%) on the MATH dataset, as mentioned several times in the Introduction, Method, and Experiments sections. For example, in the first paragraph of Experiments (Section 4.1, Page 7), we note that GPT4-Code reaches 69.69% on MATH, largely surpassing the previous SOTA result (53.90%). This demonstrates that GPT4-Code exhibits strong abilities in solving math problems. We then used the GPT4-Code's result (69.69%) as **our baseline**.
>
> To avoid misunderstanding and demonstrate our contributions more clearly, we have modified Tab. 1, Fig. 2, and Fig. 6 and adjusted expressions in many places in the paper.
>
> **In summary, our contributions are as follows**: We analyze the underlying code mechanism, develop a code-based self-verification (CSV) prompt, and introduce a weighted majority voting method. We also conduct quantitative experiments on various datasets (see detailed contributions in the Introduction). Based on your comments, we have more clearly acknowledged the high performance of the GPT-4 Code and analyzed it in depth. Additionally, we have consistently mentioned throughout the paper that our improvements are based on the GPT-4 Code.

---

> ### Author Response · Authors · 2023-11-20
> **Response to Official Review by Reviewer UL1M (2/3)**
>
> **Q3**: For example in Page 6, the authors said "Before the advent of GPT4-Code, prior frameworks (Lightman et al., 2023; Cobbe et al., 2021) depended on an external LLM to use natural language for verification and well-designed few shot example prompts. In contrast, our approach simplifies the process by relying solely on a straightforward prompt for GPT4-Code, all in a zero-shot manner." But the root reason that zero-shot is applicable is because you are using OpenAI's instruction-tuned model.
>
>
> **A3**: Thank you for your feedback. The phrase "Before the advent of GPT4-Code... in a zero-shot manner" is a bit of an overstatement. Thank you for pointing this out. As demonstrated in our experiments, the sentence was intended to highlight the distinction between GPT4-Code and other verification methodologies (Lightman et al., 2023; Cobbe et al., 2021). In hindsight, we should not have written the ambiguous sentence that makes it seem as if we are claiming the zero-shot capability as our own innovation. We have revised the statement to read: "Before the advent of GPT4-Code, prior frameworks (Lightman et al., 2023; Cobbe et al., 2021) relied on an external Large Language Model (LLM) and well-constructed few-shot prompts for natural language verification. In contrast, GPT4-Code's robust capabilities **enable our approach** to depend solely on a straightforward prompt, thereby operating in a zero-shot manner." We apologize for any confusion caused.
>
>
> Moreover, along with other works in zero-shot [1, 2, 4] and few-shot [2, 3] prompting, utilizes carefully designed prompts to tap into the latent capabilities of large language models (LLMs). This exploration not only demonstrates the potential of LLMs but also bridges their power with practical applications. In this paper, we analyze the abilities and discuss the mechanisms behind OpenAI's advanced model, GPT-4 Code, using code-constrained and CSV prompting techniques. We believe these insights are of great value to the community.
>
> In addition, We have tested our method on CodeLlama-7B and CodeLlama-34B by applying the same prompting method of CSV to these weaker models in a zero-shot manner.  The results are shown in the table below.  There are also noticeable improvements on both GSM8K and MATH.
>
> |     | GSM8K Acc (%)|MATH Acc (%) |
> |:------------------|:------------:|:-----------:|
> | CodeLlama-7B       | 17.44        | 6.56        |
> | CodeLlama-7B + CSV | 20.85        | 10.18         |
> | CodeLlama-34B      | 28.96        | 9.12        |
> | CodeLlama-34B + CSV| 37.60         | 13.36       |
>
>
>
>
> [1]. Kojima, T., Gu, S. S., Reid, M., Matsuo, Y., & Iwasawa, Y. (2022). Large language models are zero-shot reasoners. Advances in neural information processing systems, 35, 22199-22213.
>
> [2]. Chen W, Ma X, Wang X, et al. Program of thoughts prompting: Disentangling computation from reasoning for numerical reasoning tasks[J]. Transactions on Machine Learning Research.(https://openreview.net/forum?id=YfZ4ZPt8zd)
>
> [3]. Wei, J., Wang, X., Schuurmans, D., Bosma, M., Xia, F., Chi, E., ... & Zhou, D. (2022). Chain-of-thought prompting elicits reasoning in large language models. Advances in Neural Information Processing Systems, 35, 24824-24837.
>
> [4]. Yang, C., Wang, X., Lu, Y., Liu, H., Le, Q. V., Zhou, D., & Chen, X. (2023). Large language models as optimizers. arXiv preprint arXiv:2309.03409.
>
> **Q4**: In addition, it is very likely that OpenAI has already used PRM (Lightman et al., 2023) to RL tune it's new model, therefore the API you used could be based on top of Lightman et al., 2023.
>
> **A4**: We cannot confirm whether GPT-4 Code incorporates the PRM methodology (Lightman et al., 2023), a recent preprint dated May 31, 2023. The key points are as follows:
>
> 1. PRM is specifically tailored for math problem-solving, whereas GPT4-Code is a general-purpose code interpreter with robust capabilities across various domains.
> 2. The PRM study validates a subset of the MATH dataset, termed 'minival', and achieves 78.2% accuracy [2]. Our testing of GPT4-Code on the same minival dataset resulted in a noticeably lower accuracy of 70.8%.
>
> Therefore, we can speculate that GPT4-Code most likely does not exploit the PRM technique (Lightman et al., 2023). The comparison is summarized in the table below:
>
> | minival       | Accuracy  |
> |:--------------|:---------:|
> | PRM(Lightman et al., 2023) | 78.2|
> |GPT-4-Code      | 70.8            |
> | GPT4-Code + CSV (Ours)    | 75.2 |
>
>
> Furthermore, PRM utilizes a reward model to verify the accuracy of each step in a solution. In contrast, our method employs CSV prompting to encourage GPT-4 Code to generate additional steps to validate the entire solution, which is fundamentally different from the approach used in PRM (Lightman et al., 2023).
>
>
>
>
> [1]. Lightman, Hunter, et al. "Let's Verify Step by Step." arXiv preprint arXiv:2305.20050 (2023).
>
> [2]. https://paperswithcode.com/sota/math-word-problem-solving-on-math-minival

---

> ### Author Response · Authors · 2023-11-20
> **Response to Official Review by Reviewer UL1M (3/3)**
>
> *Hope our above responses are helpful to address your concerns. If you have any questions, please let us know. Thanks!*

---

### Official Review · Reviewer_boig · 2023-11-01

**Soundness:** 3 good
**Presentation:** 4 excellent
**Contribution:** 4 excellent
**Rating:** 8
**Confidence:** 5

**Summary:**

This paper presents a prompting based self-verification technique to improve GPT-4 CodeIntepreter model's performance on math tasks. The key ideas are: (1) by prompting the model to perform code-based self-verification, the model improves its performance, and (2) taking advantage of self-verification result for weighted-voting, the model further improves its performance.

The paper is inline with existing lines of work of leveraging model's self-debugging ability to improve math reasoning ability. The experiments showed that (1) by leveraging multiple invocation of code call, GPT-4 Code achieve higher performance comparing to NL or 1-code call variances, (2) self-verification provides the opportunity for the model to discover its own error thus to fix it, and (3) voting solutions based on self-verification results achieves better performance over vallina voting scheme.

**Strengths:**

This paper is timely. It's a novel approach of leveraging code self-verification to improve math reasoning ability. It's simplicity means it can be easily used by GPT-4 users, and the thorough study provides solid evidence of adopting this technique.

Concretely, the paper's strengths include:
1. Novel technique that combines self-debugging and test generation for math reasoning tasks.
2. Quite complete study that highlights effectiveness of the technique.
3. Weighted voting technique is quite unique, and it can possibly used in other self-debugging techniques beyond Math reasoning tasks.

**Weaknesses:**

The paper lacks some insights into the quality of self-verification results, and how that matters for model performance.
1. The paper can potentially dive deeper into analysis of consistency between verification process and NL reasoning process as well as output correctness. As shown in prior work like CodeT, some self-generated test-cases or verification code can either be wrong or inconsistent, but it may or may not affect model output quality. Guessing from the paper's results on weighted voting versus simple voting, such inconsistency exists and they could benefit model performance (or affect model performance if we simply reject such answers). I would suggest the authors perform some qualitative analysis to dive into this problem.

1b. Some deeper qualitative analysis into what types of verification code are generated would also be helpful.

2.The paper considers both sampling and sequential self-repair. It would be great if the authors can analyze the tradeoff between the depths of self-repair (e.g., if self-verification continues to fail, how much can the model benefit from continuing self-repair until verification succeed) verse breaths of self-verification (e.g., simply repair once, but generate multiple samples to do weighted sampling).

3. The paper plots are visually misleading: Figure 2 accuracy numbers should all start from 0 as opposed to 60 in figure 2a and 40 in figure 2b. Otherwise the improvement looks like 10x as opposed to 6% comparing prompts 1 and 2. Similarly for other figures.

**Questions:**

I think the paper lacks a qualitative analysis of the self-verification code and analysis of consistency between answer and verification code. Otherwise a great paper showing the ability of self-verification.

---

> ### Author Response · Authors · 2023-11-20
> **Response to Official Review by Reviewer boig (1/2)**
>
> We sincerely thank you for your comments as well as your appreciation of our work. Our responses to your concerns are demonstrated below:
>
> **Q1**: The paper can potentially dive deeper into the analysis of consistency between verification process and NL reasoning process as well as output correctness.I would suggest the authors perform some qualitative analysis to dive into this problem.
>
> **A1**: Thank you for suggesting this interesting analysis. We did some qualitative analysis and provided examples of the consistency between the correctness of the solution process and the verification process. There are four cases:
>
>
>
>
> The verification process is primarily consistent with the NL reasoning process. The model would start the verification by reevaluating the problem and providing a general plan for verification in natural language. Then, it would generate the code based on this plan. Finally, it would output the verification result after analyzing the output of the verification. The verification is a seamless continuation of the NL reasoning process.
>
> To further analyze the consistency between the verification process, NL reasoning process, and output correctness, we have collected examples of four kinds listed in the table below. The reasoning process is correct if the answer before the verification is correct. The verification process is correct if the verification process itself is correct. We provide the examples with four anonymous links.
>
>
> |Case| Reasoning Process | Verification Process| Anonymous Link|
> |:-------------------:|:-------------------:|:-----------:|:---------------|
> | **1** | Correct | Correct     |  https://anonymous.4open.science/r/qualitative-analysis-of-consistency-between-verification-process-and-reasoning-process-0C64/examples/example1.md|
> | **2** |  Correct   | Wrong      | https://anonymous.4open.science/r/qualitative-analysis-of-consistency-between-verification-process-and-reasoning-process-0C64/examples/example2.md|
> | **3** |  Wrong   | Correct      | https://anonymous.4open.science/r/qualitative-analysis-of-consistency-between-verification-process-and-reasoning-process-0C64/examples/example3.md|
> | **4** |  Wrong   | Wrong      | https://anonymous.4open.science/r/qualitative-analysis-of-consistency-between-verification-process-and-reasoning-process-0C64/examples/example4.md|
>
> **1.**  When both the reasoning and verification processes are correct, the first verification successfully validates the solution, ensuring consistency with the answer and the natural language (NL) reasoning process.
>
> **2.**  If the reasoning is correct but the verification is erroneous, the first time verification fails to recognize the correct solution.
>
> **3.**  If the reasoning is incorrect but the verification is accurate, the verification process amends the erroneous reasoning, leading to a correct answer.
>
> **4.**  If both the reasoning and verification processes are incorrect, the verification is unable to rectify the flawed solution.
>
>
> **Q2**: Some deeper qualitative analysis into what types of verification code are generated would also be helpful.
>
> **A2**: Upon inspection of the generated solutions, we have found the following four types of verification code:
>
> 1. **Substitution**. The model substitutes the answer back to the question to check if it fits all the conditions. By inspecting the solutions we randomly sampled from the results, we found that this is the most common type of verification.
>
> 2. **Alternative Method**. The model uses a different method to solve the problem and check if it reaches the same final answer.
>
> 3. **Double Checking**. The model double-checks each part of the solution to see if they are correct.
>
> 4. **Approximation**. The model computes an approximation of the answer. This method is usually used when there is infinity involved in the problems. The model uses a large number to approximate the infinity to check if the result is close to the answer.
>
> Corresponding examples are presented in an anonymous link [1]
>
> [1] https://anonymous.4open.science/r/CSV_rebuttal_examples-7D47/verify_type_code.md

---

> > ### Comment · Reviewer_boig · 2023-11-21
> >
> > Thanks for diving deeper into qualitative analysis; these results should be part of the paper.
> >
> > Furthermore, in the revision phase, if possible, I would suggest authors conduct qualitative analysis in a more systematic way: e.g., randomly select 50 tasks and inspect them to check how often they fall into different categories. This would be a good setup for motivating future research.

---

> ### Author Response · Authors · 2023-11-20
> **Response to Official Review by Reviewer boig (2/2)**
>
> **Q3**:It would be great if the authors can analyze the tradeoff between the depths of self-repair (e.g., if self-verification continues to fail, how much can the model benefit from continuing self-repair until verification succeed) verse breaths of self-verification (e.g., simply repair once, but generate multiple samples to do weighted sampling).
>
> **A3**: Thank you for your constructive suggestions. Due to the limited window size of GPT4-Code, we cannot guarantee that the model will continue self-repairing until verification is successful. Consequently, we can only compare unlimited self-repair with majority voting of limited self-repair.
>
>  Majority voting with limited self-repair has a higher accuracy than unlimited self-repair. However, it also induces extra cost. To analyze the tradeoff between the depth of self-repair and the breadth of self-verification, we compare the results of two scenarios: one limiting the depth of self-repair to 1 and another allowing unlimited self-repair. The results are shown below:
>
> |           | MATH Acc (%) |
> |:----------:|:----------:|
> | Unlimited (average depth 1.22) | 73.54 |
> | self-repair once + voting (k=3) | 77.02 |
>
> As shown above, when self-repair is limited to one attempt, voting reaches an accuracy of 77.02, which surpasses the accuracy of unlimited self-repair without voting by 3.48 percentage points. However, in the case of voting, the number of solutions generated for each problem is three. This is significantly higher than the average number of attempts in unlimited self-repair, which is 1.22.
>
> **Q4**: The paper plots are visually misleading: Figures accuracy numbers should all start from 0.
>
> **A4**: We acknowledge your concern about the starting points of the y-axis in the accuracy graphs, which may unintentionally exaggerate the improvements. To minimize potential confusion, we have revised Figures 2a, 2b, and 6a to start from 0. We have also adjusted Table 1 by changing the baseline of voting to GPT4-Code + Voting. Further adjustments can be made if you have more suggestions.
>
> *Hope our above responses are helpful to address your concerns. If you have further questions, please let us know. Thanks!*

---

> ### Author Response · Authors · 2023-11-22
> **Response to Reviewer boig**
>
> Thank you for your constructive suggestions. Based on your suggestions, we have revisited and addressed Questions 1 and 2 in a more systematic manner.
>
> **Q1**: The paper can potentially dive deeper into the analysis of consistency between verification process and NL reasoning process as well as output correctness. I would suggest the authors perform some qualitative analysis to dive into this problem.
>
> **A1**:  As we mentioned in our previous answer, we collected examples of four kinds, as listed in the table below. We have since randomly selected 100 samples and manually checked which kind they belong to. The percentage of samples belonging to the four kinds is 76%, 5%, 7%, and 12% respectively.
>
> |Case| Reasoning Process | Verification Process| Percentage (%) |
> |:-------------------:|:-------------------:|:-----------:|:---------------|
> | 1 | Correct | Correct     |  76 |
> | 2 |  Correct   | Wrong      | 5 |
> | 3 |  Wrong   | Correct      | 7 |
> | 4 |  Wrong   | Wrong      | 12 |
>
> This shows that in most of the samples, **the reasoning process and the verification process are consistent with each other**. To better visualize the results, we present a **pie chart showing the distribution** in an anonymous link [1]. We have updated the paper and added the chart and some analysis to Appendix I (page 24).
>
> [1] https://anonymous.4open.science/r/verify_consistency-F602/README.md
>
> **Q2**: Some deeper qualitative analysis into what types of verification code are generated would also be helpful.
>
> To further analyze the types of verification code generated, we randomly sampled 50 problems from MATH as well as their solutions generated with CSV. We counted four types of verification code: **Substitution, Alternative Method, Double Checking, and Approximation**. We have already explained what these four types mean in our previous answer. As shown in the table below, the percentage of samples belonging to the four types is 50%, 22%, 18%, and 10%, respectively.
>
> |Case| Verification Type | Percentage (%) |
> |:-------------------|:-------------------|:---------------|
> | 1 | Substitution |  50 |
> | 2 | Alternative Method  | 22 |
> | 3 | Double Checking  | 18 |
> | 4 | Approximation  | 10 |
>
> Among the verification types, **Substitution and Alternative Methods** are the most common, comprising 50% and 22% respectively. We also present a **pie chart showing this distribution** through an anonymous link [1]. We have updated the paper to include the chart and some analysis in Appendix J (pages 24 and 25).
>
> [1] https://anonymous.4open.science/r/CSV_rebuttal_examples-FE7D/verify_type_code.md
>
> Thank you again for your valuable suggestions.

---

> > ### Comment · Reviewer_boig · 2023-11-23
> >
> > Thanks for adding these results. I suggest include these materials in the main paper.
> >
> > As Reviewer UL1M mentioned, "technical contribution" of the paper doesn't really shine. Thus, I view this paper more of a study/analysis paper that would benefit GPT-4 Code users. For a solid study paper, both qualitative and quantitative results should be included in the main paper.
> >
> > Thanks for the revision effort. I remain positive of the paper.

---

> > > ### Author Response · Authors · 2023-11-23
> > > **Thanks for your response.**
> > >
> > > Dear Reviewer boig,
> > >
> > > Thank you for acknowledging our response and efforts! We appreciate your valuable time and constructive comments.
> > >
> > > Sincerely,
> > > Authors

---

### Official Review · Reviewer_qqMb · 2023-11-01

**Soundness:** 2 fair
**Presentation:** 3 good
**Contribution:** 2 fair
**Rating:** 6
**Confidence:** 4

**Summary:**

This paper studies the effectiveness of code writing and self-validation for improving the reasoning capability of large language models (LLMs). Given that GPT-4 can generate Python programs, instead of providing step-by-step reasoning entirely in natural language text, this paper suggests a new prompt that encourages GPT-4 to generate both text and Python code snippets as well as a self-validation in the end. Additionally, a weighted majority voting is proposed to aggregate multiple runs or samples. The evaluation on three math datasets shows that GPT-4 with proposed prompting techniques significantly outperforms the previous state-of-the-art results.

**Strengths:**

- Like many recent works, this work provides strong evidence that prompting LLMs in a proper way could significantly influence the performance
- This work also makes another natural but still interesting finding -- the frequency of code usage has a strong correlation with the accuracy of the final answer. The rationality is that executing code is more accurate/reliable than performing reasoning through natural language text.
- This work presents the new state-of-the-art results on three datasets (i.e., MATH, GSM8K, MMLU-Math).

**Weaknesses:**

- In terms of improving LLMs, the ideas of code writing, self-validating, and majority voting have been already explored in recent literature. The novelty of this work seems a simple combination of all three ideas together.

- Only one particular and proprietary LLM (i.e., GPT-4) is used for evaluation. Whether the results and findings reported in this work may generalize to other publicly available LLMs (e.g., Llama 2) or relatively smaller language models is unclear.

- Another similar concern is that only one specific kind of benchmark (i.e., grade school math problems) is used, thus the proposed prompt might be overfitting to simple math problems.  For the chosen benchmarks, improvements due to coding and CSV on GSM8K and MMLU-Math are already fairly small (i.e., 2-4%).

- The improvement seems largely due to the superior capability of GPT-4. For instance, GPT-4 itself can outperform the state-of-the-art by a large margin without any sophisticated prompting and majority voting. It is already well-known that even a simple prompt could dramatically influence performance (see ). The particular finding of GPT-4 on the MATH benchmark is not very surprising.

**Questions:**

Would the same idea improve other LLMs like Llama 2? Or what factors may prevent the same idea from improving other LLMs?

In Table 3, the last row only mentions _CSV_ (but not Voting), is it a typo? If not, why voting is not enabled?

---

> ### Author Response · Authors · 2023-11-20
> **Response to Official Review by Reviewer qqMb (1/2)**
>
> Thank you for your valuable feedback. We would like to address your concerns from the following aspects:
>
> **Q1**: In terms of improving LLMs, the ideas of code writing, self-validating, and majority voting have been already explored in recent literature. The novelty of this work seems a simple combination of all three ideas together.
>
> **A1**: Thank you for your valuable feedback. We would like to highlight the novelty of our work from the following four aspects.
>
> First of all, we are the first to discover the important role code plays in GPT-4 Code Interpreter’s high performance in solving math problems via prompt engineering. GPT-4 Code Interpreter uses solutions interleaved with natural language, code, and execution results to solve math problems, possessing the ability to analyze execution results and self-repair. To the best of our knowledge, this is the first quantitative experimental analysis of this interesting phenomenon.
>
> Secondly, our method employs zero-shot prompting for automatic code-based self-verification, exploring GPT-4 Code Interpreter’s ability to use code multiple times, interleaved with natural language reasoning. Existing work on self-validation uses additional validator mechanisms ([3], [4], [5]), while our method employs zero-shot prompting for automatic self-verification. Additionally, our work uses code to verify the solution of math problems, which is different from existing work as well.
>
> Thirdly, we employ weighted voting based on the results from the proposed code-based self-verification, thereby improving the performance of existing majority voting, as demonstrated in Figure 6. In contrast, most existing works use majority voting. This verification-guided weighted majority voting has never been studied before.
>
> Fourthly, our paper focuses on the potential of using code multiple times, **interleaved with natural language reasoning**, while current methods using code, such as PAL [1] and PoT [2], generally only use code once.
>
> In summary,  although there are existing language models on code writing, self-validating, and majority voting, our method is notably different from theirs. We believe our approach of quantitative analysis and enhancing GPT4-Code with self-verification provides valuable and novel insights into the current top Large Language Models (LLMs).
>
> **Q2**: Another similar concern is that only one specific kind of benchmark (i.e., grade school math problems) is used, thus the proposed prompt might be overfitting to simple math problems. For the chosen benchmarks, improvements due to coding and CSV on GSM8K and MMLU-Math are already fairly small (i.e., 2-4%).
>
> **A2**： Thank you for your valuable feedback regarding the easier benchmarks.
>
> Regarding the concern that only one specific kind of benchmark (i.e., grade school math problems) is used, the MATH dataset consists of challenging competition-level problems of multiple math fields, including algebra, number theory, geometry, precalculus and so on [6]. Many of these problems are challenging even for college students, and the previous SOTA on this dataset is 53.9%, by the PHP method used on GPT4. GPT4-Code improves this score to 69.69%. Additionally, it is essential to note that the proposed prompt significantly improves the MATH dataset (69.69% -> 84.32%), demonstrating great potential for solving difficult math problems.
>
> As for other relatively easier datasets, such as GSM8K and MMLU-Math, our proposed CSV + VW-Voting method demonstrates a smaller yet noticeable improvement compared to the MATH dataset. This relatively smaller improvement is not surprising, given these datasets' already quite high baseline performance. To fully appreciate the improvement induced by our method, we would like to calculate Relative error reduction **Ratio**, which is defined as $Ratio = \frac{Err_{baseline}-Err_{our}}{Err_{baseline}}$, where $Err = 1 - Acc$.
>
> The results are demonstrated in the tables below. As can be seen, on GSM8K, our method reduces the Err by 4.1 percentage points, resulting in a Ratio of 57.7%. Experiments on MMLU-Math and MATH yield Ratios of 56.0% and 48.3%, respectively, showing a pattern similar to that observed with GSM8K.
>
> | GSM8K                       | Err (%)| Ratio (\%) |
> |:---------------------------|:------:|:---------:|
> | GPT4-Code (baseline)        | 7.1    |  --       |
> | GPT4-Code + CSV + VW-Voting | 3.0    | 57.7      |
>
> | MMLU-Math                   | Err (%)| Ratio (\%) |
> |:---------------------------|:------:|:---------:|
> | GPT4-Code (baseline)        | 12.5   |  --       |
> | GPT4-Code + CSV + VW-Voting | 5.5    | 56.0      |
>
> | MATH                        | Err (%)| Ratio (\%) |
> |:---------------------------|:------:|:---------:|
> | GPT4-Code (baseline)        | 30.31  |  --       |
> | GPT4-Code + CSV + VW-Voting | 15.68  | 48.3      |

---

> ### Author Response · Authors · 2023-11-20
> **Response to Official Review by Reviewer qqMb (2/2)**
>
> **Q3**: Only one particular and proprietary LLM (i.e., GPT-4) is used for evaluation. Whether the results and findings reported in this work may generalize to other publicly available LLMs (e.g., Llama 2) or relatively smaller language models is unclear.
>
> **A3**: We have tested our method on CodeLlama 7B and 34B by applying the same prompting of CSV to these weaker models in a zero-shot manner. The results are shown in the table below. As can be seen, there are also noticeable improvements on both GSM8K and MATH, though the accuracy is much lower compared to GPT4-Code.
>
> |                    | GSM8K Acc (%)|MATH Acc (%) |
> |:------------------|:------------:|:-----------:|
> | CodeLlama-7B       | 17.44        | 6.56        |
> | CodeLlama-7B + CSV | 20.85        | 10.18         |
> | CodeLlama-34B      | 28.96        | 9.12        |
> | CodeLlama-34B + CSV| 37.6         | 13.36       |
>
> **Q4**: The improvement seems largely due to the superior capability of GPT-4. For instance, GPT-4 itself can outperform the state-of-the-art by a large margin without any sophisticated prompting and majority voting. It is already well-known that even a simple prompt could dramatically influence performance. The particular finding of GPT-4 on the MATH benchmark is not very surprising.
>
> **A4**:  We agree that “even a simple prompt could dramatically influence performance.” However, how to perform “prompt engineering” to discover the appropriate prompt for making GPT-4 Code improve itself is not straightforward. We conducted a pioneering analysis to understand the key factor contributing to GPT-4 Code’s high performance on math problems and we propose the code-based self-verification prompt based on our analysis. In addition, we also found the new prompt is generalizable and can be used for CodeLlama to improve its math-solving accuracy as well. It is non-trivial to figure out such a generalized prompt.
>
> To make our contribution clearer, we have added the baseline of **GPT4-Code + Voting**, which is the result of naive majority voting and GPT4-Code without CSV. This baseline is 79.88%, which is shown in Table 1. All the methods that involve voting have been adjusted to be comparable to this baseline. We hope this will help readers better understand the contributions of this paper. We have listed the related results on  MATH below. Our method has a stable and noticeable effect based on the GPT4-Code baseline.
>
> |                      | MATH Acc (%) | Improvement (%) |
> |:--------------------|:------------:|:---------------:|
> | GPT4-Code (baseline) | 69.69        | --              |
> | GPT4-Code + CSV      | 73.54        | +3.85           |
>
> |                                    | MATH Acc (%)| Improvement (%)|
> |:----------------------------------|:-----------:|:--------------:|
> | GPT4-Code + Voting (k=16, baseline)| 79.88       | --             |
> | GPT4-Code + CSV + Voting (k=16)    | 83.54       | +3.66          |
> | GPT4-Code + CSV + VW-Voting (k=16) | 84.3        | +4.44          |
>
>
> **Q5**: In Table 3, the last row only mentions CSV (but not Voting), is it a typo? If not, why voting is not enabled?
>
> **A5**: Thank you for pointing this out. The absence of voting results was not a typo. Initially, due to our GPT-4 quota limits at the time of submission, we did not run MMLU-Math multiple times, which is necessary to obtain voting results. However, we have since conducted the experiment. We have updated Table 3 to include the outcome for GPT4-Code + CSV + VW-Voting on MMLU-Math, which achieves 94.5\% accuracy.
>
> [1] Luyu Gao, Aman Madaan, Shuyan Zhou, Uri Alon, Pengfei Liu, Yiming Yang, Jamie Callan, and Graham Neubig. PAL: Program-aided language models. In International Conference on Machine Learning, pp. 10764–10799. PMLR, 2023.
>
> [2] Wenhu Chen, Xueguang Ma, Xinyi Wang, and William W. Cohen. Program of thoughts prompting: Disentangling computation from reasoning for numerical reasoning tasks, 2022.
>
> [3] Karl Cobbe, Vineet Kosaraju, Mohammad Bavarian, Mark Chen, Heewoo Jun, Lukasz Kaiser, Matthias Plappert, Jerry Tworek, Jacob Hilton, Reiichiro Nakano, Christopher Hesse, and John Schulman. Training verifiers to solve math word problems. arXiv preprint arXiv:2110.14168, 2021.
>
> [4] Hunter Lightman, Vineet Kosaraju, Yura Burda, Harri Edwards, Bowen Baker, Teddy Lee, Jan Leike, John Schulman, Ilya Sutskever, and Karl Cobbe. Let’s verify step by step. arXiv preprint arXiv:2305.20050, 2023.
>
> [5] Yixuan Weng, Minjun Zhu, Fei Xia, Bin Li, Shizhu He, Kang Liu, and Jun Zhao. Large language models are better reasoners with self-verification, 2023.
>
> [6] Dan Hendrycks, Collin Burns, Saurav Kadavath, Akul Arora, Steven Basart, Eric Tang, Dawn Song, and Jacob Steinhardt. Measuring mathematical problem solving with the math dataset. arXiv preprint arXiv:2103.03874, 2021.
>
>
> *Hope our above responses are helpful to address your concerns. If you have further questions, please let us know. Thanks!*

---

> > ### Author Response · Authors · 2023-11-22
> > **Looking forward to your reply**
> >
> > Dear Reviewer qqMb,
> >
> > We hope this message finds you well. As the author-reviewer discussion deadline approaches, we respectfully seek your confirmation on the adequacy of our rebuttal in addressing the concerns raised in your review.
> >
> > We really appreciate the substantial time and effort you have committed to reviewing our work and are grateful for the additional insights. Your comments have been very helpful in refining our project.
> >
> > Thank you once again for your valuable perspectives. We eagerly await your further guidance.
> >
> > Sincerely,
> > Authors

---

> > > ### Comment · Reviewer_qqMb · 2023-11-22
> > >
> > > Thank you for sharing some new results. I am generally concerned about the paradigm of combining all kinds of tricks on top of a proprietary model. The obvious danger is that the finding is very likely tailored to that specific model.  With that said, the new result about CodeLlama certainly adds some value. My confusion is that these numbers are way lower than the work Llama-2(70B) (Touvron et al., 2023) on the same task, which confirms my concern that CSV does not work well for other LLMs since it could not even fill the gap between variances of very similar public LLMs like LLama. Given the new result provides some excitement but also a little bit of concern, I will keep my original rating. Regardless of the outcome, I do appreciate the authors' effort.

---

> ### Author Response · Authors · 2023-11-22
> **Thanks for your reply!**
>
> Dear Reviewer qqMb,
>
> Thank you for your response and additional constructive advice! We hope our further response can address your concerns.
>
> Firstly, the Llama2-70B model is **significantly larger** than CodeLlama-34B. *Comparing **34B** model with a **70B** one is not a fair assessment*. Our paper focuses on code-based models, and since Llama2 is weaker in code generation capabilities, we did not choose it as our base model. Moreover, we cannot access a same-scale (70B) CodeLlama model. In fact, CSV's performance on CodeLlama is actually superior to that of a Llama2 model of the same size. This is evident in the table provided, where our performance on CodeLlama-34B exceeds that of Llama2-34B on the GSM8K and MATH benchmarks, with gains of 4.85 and 9.16, respectively.
>
> Secondly, all of our performance metrics are based on zero-shot prompting, maintaining consistency with the main experiments in our paper. This contrasts with the settings for Llama2, as they report: "We report the average of the GSM8K (8 shot)  and MATH (4 shot) benchmarks at top 1." Therefore, the comparison settings are not equivalent.
>
> Thirdly, when comparing Codellama-34B with and without CSV, we observe a significant performance enhancement: *an **8.64%** gain on the MATH benchmark and a **4.24%** gain on GSM8K, which is similar to the findings on GPT4-Code.* These improvements are substantial and cannot be merely attributed to variance without the use of CSV. The detailed results are as follows:
>
> | model         | size |  GSM8K      | few-shot| MATH        | few-shot |
> |:-------------------|:-----------|:--------------|:-----------|:--------------|:--------------|
> | Llama2        | 7B   | 14.60          | **8-shot**    | 2.50          | **4-shot**    |
> | CodeLlama     | 7B   | 17.44         | zero shot | 6.56          | zero shot |
> | **CodeLlama+CSV** | **7B**   | **20.85**         | **zero shot** | **10.18**         | **zero shot** |
> | Llama2        | 34B  | 42.20         | **8-shot**    | 6.24          | **4-shot**    |
> | Llama2        | 34B  | 32.75         | zero shot | 4.20          | zero shot |
> | CodeLlama     | 34B  | 28.96         | zero shot | 9.12          | zero shot |
> |**CodeLlama+CSV** | **34B**  | **37.60 (+8.64)** | **zero shot** | **13.36 (+4.24)** | **zero shot** |
>
>
> *We appreciate your acknowledgment of our efforts and hope this response clarifies your concerns.*
>
> Sincerely,
>
> Authors

---

> ### Comment · Reviewer_qqMb · 2023-11-22
>
> With this very recent new result, I am now positive about this work. They should be included in the final revision. Adding a setting of LLama2+CSV would make the evaluation more systematic and convincing (which is not urgent and can be done at a later time).
>
> Given these new results, having GPT-4 in the title seems not appropriate, as now the contribution is not limited to GPT-4.

---

> ### Author Response · Authors · 2023-11-23
> **Thanks for your response.**
>
> Dear Reviewer qqMb,
>
> Thank you for your response. We appreciate your valuable time and constructive comments.
>
> We plan to add the results of CSV on open-source models, including Llama-2 + CSV, to our revised paper. Additionally, based on your suggestions, we are considering changing our title to “Solving Challenging Math Word Problems using Large Language Models with Code-based Self-verification” for the revised version.
>
> Thank you again for your constructive suggestions.
>
> Sincerely,
> Authors

---

### Official Review · Reviewer_iedh · 2023-11-08

**Soundness:** 3 good
**Presentation:** 3 good
**Contribution:** 2 fair
**Rating:** 8
**Confidence:** 3

**Summary:**

This paper proposes a zero-shot prompting strategy for LLMs that can execute code during a completion (e.g. GPT-4 Code Interpreter). The strategy, referred to as CSV (code-based self-verification), is to use the following zero-shot prompt: "Solve the problem using code interpreter step by step, even in every sub-step. And following your answer, please verify it using code interpreter by yourself." They find that this prompt boosts performance on several math word problem datasets. The authors then present a variation on majority voting based on weighting samples that were verified as True more heavily. The authors explore various ablations/variations, and also find a strong correlation between amount of code use and accuracy in general (ablations that can only use their code interpreter zero or one times perform worse).

**Strengths:**

- The results on correlations between code usage frequency and accuracy are a nice analysis, and breaking it down by difficulty level of problem is useful (e.g. at higher difficulty levels amount of code usage seems to matter more)
- The CSV setup itself is quite simple, which is a good thing – "Solve the problem using code interpreter step by step, even in every sub-step. And following your answer, please verify it using code interpreter by yourself." A simple way of getting a decent boost in performance from zero-shot prompting is a fairly useful contribution. The boost of 3.85% accuracy on the MATH dataset from using CSV over the base prompt is solid – that's a reasonable gain.
- The Verification-guided weighted majority voting setup is also quite simple (just a weighted majority vote with human-picked parameters) which is good. The weighting seems to provide a boost of around +1% accuracy over a standard majority voting approach (at 16 samples, MATH dataset, per Fig 6b).
- The precision-recall-accuracy analysis and also analysis of weighted majority vs naive majority are good to see, I was preparing to suggest that, and I'm glad it was done.

**Weaknesses:**

Overall I'm marginally below acceptance on this paper, but would certainly consider raising my score if the following points are well addressed in revisions/rebuttal.
- As-is the current paper has only one row in Table 1 that uses 16 samples with GPT4-Code, so the +14.63% improvement is a comparison between a 1 sample and 16 sample method. This can be a bit misleading to readers (or at least it was to me) since much of that improvement comes from using additional samples, which is something that could also be done through prior work via naive majority voting (which is actually evaluated later in the paper, in Fig 6b). Therefore, Table 1 should include a line for **GPT4-Code + CSV + Majority Voting** (which does about 1% worse than weighted voting, per Fig 6b).
- For the same reasons as in the previous comment, adding a line to Table 1 with **GPT4-Code + Majority Voting (no CSV)**, an ablation that done with neither CSV nor weighted voting, would be an informative baseline to include. These two baselines would considerably strengthen the analysis.
- There doesn't seem to be a **GPT4-Code + CSV** row in Table 2, nor a **GPT4-Code + CSV + Voting** row in Table 3. This use of different setups on different datasets is confusing – an explanation should be given, or preferably both should be added. Having both of these novel methods from the paper evaluated on all 3 datasets would strengthen the paper.
- Figure 6a is quite confusing to me:
    - Which dataset is used? I believe it's MATH overall dataset, but this should be in the caption and optionally also main text
    - I don't understand why there are 5 separate paths graphed, given that there's nothing special differentiating path 0 from path 1, etc. The different path indices are just IID samples, so of course when you calculate precision/accuracy/recall averaged over a whole dataset of problems, all the histogram bars will look the same for different sample path indices. So having 5 separate paths doesn't seem to add anything – why not just collapse them all into a single graph with three bars (accuracy, precision, recall)?

minor:
- Consider weakening the statement "Each line in Fig. 5 has an obvious trend of going upwards, proving that the increase of Code Usage Frequency induces a general improvement in accuracy": *Proves* is a strong word there, when it's just a nice correlation (not causal – there are many ways of including meaningless code in an output that wouldn't not induce an improvement in accuracy). Just state it as a correlation and that's good.

**Questions:**

- Can you address or respond to the main 4 points made in the weaknesses section?
    - The majority voting ablation addition to Table 1
    - The majority voting + no CSV ablation addition to Table 1
    - The missing rows of Table 2 and Table 3
    - The confusion around Fig 6a

More minor questions:
- In Fig 2 why are some of the red dots different sizes given that the prompt says no code is allowed – does that mean in some cases it's using code anyways?
- "Moreover, we observed a decline in accuracy for 4 of the seven subtopics, indicating that relying solely on natural language self-verification can not only compromise accuracy but also negatively impact performance" What does this line mean – aren't accuracy and performance the same thing?
- Do you have a sense for why the natural language verification row in Table 2 can fairly significantly *decrease* performance (by 2-3% in some domains)?

---

> ### Author Response · Authors · 2023-11-20
> **Response to Official Review by Reviewer iedh (1/2)**
>
> Thank you for your valuable feedback. It is true that there are a few flaws in the tables and figures in the paper.  We have revised the paper based on your suggestions, as detailed below:
>
>
> **Q1**: As-is the current paper has only one row in Table 1 that uses 16 samples with GPT4-Code, so the +14.63% improvement is a comparison between a 1 sample and 16 sample method. much of that improvement comes from using additional samples, which is something that could also be done through prior work via naive majority voting (which is actually evaluated later in the paper, in Fig 6b). Therefore, Table 1 should include a line for GPT4-Code + CSV + Majority Voting (which does about 1% worse than weighted voting, per Fig 6b).
>
> **A1**: Thank you for pointing this out. In Table 1, we have added the line for **GPT4-Code + CSV + Voting**, with an accuracy of 83.54%. We also added the line for **GPT4-Code + Voting**, with an accuracy of 79.88%. Here, Voting is an abbreviation for Naive Majority Voting. GPT4-Code + Voting improved the baseline by 10.19%. The fact that GPT4-Code + Voting can improve the baseline by 10.19% demonstrates that GPT4-Code generates diverse and accurate solutions. Additionally, the fact that GPT4-Code + CSV + Voting is higher than GPT4-Code + Voting by 3.66% is consistent with the improvement from GPT4-Code to GPT4-Code + CSV, which is 3.85%.
>
> Also, we have noted that using the GPT4-Code result as the baseline might be misleading. Therefore, we have revised Table 1 to use GPT4-Code + Voting as the baseline for voting-related results instead.
>
> |                     | MATH Acc (%)| Improvement (%)|
> |:-------------------|:-----------:|:--------------:|
> | GPT4-Code (baseline)| 69.69       | --             |
> | GPT4-Code + CSV     | 73.54       | +3.85          |
>
> |  | MATH Acc (%)                    | Improvement (%)|
> |:----------------------------------|:--------------:|:-----:|
> | GPT4-Code + Voting (k=16, baseline)| 79.88          | --    |
> | GPT4-Code + CSV + Voting (k=16)    | 83.54          | +3.66 |
> | GPT4-Code + CSV + VW-Voting (k=16) | 84.32           | +4.44 |
>
> The tables above only list the accuracy across all the subjects. For the result of each individual subject, see Table 1 in the paper.
>
>
>
> **Q2**: There doesn't seem to be a GPT4-Code + CSV row in Table 2, nor a GPT4-Code + CSV + Voting row in Table 3. This use of different setups on different datasets is confusing – an explanation should be given, or preferably both should be added.
>
> **A2**: Thank you for pointing this out. Due to our GPT-4 quota limits at the time of submission, we did not initially run MMLU-Math multiple times. However, we have since conducted the suggested experiment. The omission of the line for **GPT4-Code + CSV** on GSM8K was an oversight, which has now been corrected with the inclusion of the corresponding result, 94.5. Also, in Table 3, we have included the result for **GPT4-Code + CSV + Voting** on MMLU-Math, which is 94.5.
>
> **Q3**: Figure 6a is quite confusing. Which dataset is used? why there are 5 separate paths graphed?
>
> In Figure 6a, the five paths graphed results of five instances of the same experiment, originally intended to demonstrate the robustness of the performance. The dataset is MATH. However, we acknowledge that the graph may be confusing to readers. We have simplified the figure to show only the average of the five instances in the revised Figure 6a, enhancing its clarity. We also changed the caption to "(a) The precision, recall, and accuracy on the MATH dataset. (b) The Acc on MATH in response to the number of sampled reasoning paths when the weight is set to different values" for clarity.
>
> **Q4**: Consider weakening the statement "Each line in Fig. 5 has an obvious trend of going upwards, proving that the increase of Code Usage Frequency induces a general improvement in accuracy": Proves is a strong word there, when it's just a nice correlation
>
> **A4**: Thanks for your reminder. We have modified the sentence to "Each line in Fig. 5 has an obvious upward trend, suggesting a possible positive correlation between Code Usage Frequency and accuracy".

---

> ### Author Response · Authors · 2023-11-20
> **Response to Official Review by Reviewer iedh (2/2)**
>
> **Q5**: In Fig 2 why are some of the red dots different sizes given that the prompt says no code is allowed – does that mean in some cases it's using code anyways?
>
> **A5**: In Figure 2, The different sizes of the red dots indicate that, in some cases, the model used code anyway. The Code Usage Frequency the red dots indicate are **0.025, 0.040, 0.046, 0.038, and 0.106** respectively, which is very small, so it doesn't pose a serious problem to the analysis. As we stated in section 3.1, "Prompt 1 results in almost negligible code usage". However, the prompt cannot completely stop the model from using code, especially when the problem is very difficult. To avoid confusion, we added the sentence "The red points denoting Prompt 1 show that the model still occasionally uses code, especially when the problem is very difficult. However, even then, the Code Usage Frequency is negligible" in the caption.
>
> **Q6**: "Moreover, we observed a decline in accuracy for 4 of the seven subtopics, indicating that relying solely on natural language self-verification can not only compromise accuracy but also negatively impact performance" What does this line mean – aren't accuracy and performance the same thing?
>
> **A6**: Thank you for pointing out this poorly written sentence.  The statement is indeed a confused expression. We originally intended to express that using natural language is generally less accurate compared to using code-based verification, thereby reducing the overall accuracy, also known as the performance. We have revised it to "Moreover, we observed a decline in accuracy of four of the seven subtopics, indicating that relying solely on natural language self-verification is less reliable than using code-based self-verification, which appears to have a negative impact on the accuracy." This revised sentence also presents our hypothesis regarding why natural language verification, as illustrated in Table 2, resulted in a significant decrease in performance.
>
> **Q7**: Do you have a sense for why the natural language verification row in Table 2 can fairly significantly decrease performance (by 2-3% in some domains)?
>
> **A7**: The decreased performance might be because natural language verification is less reliable than code-based verification, as natural language verification is prone to hallucinations, while code is precise and accurate. Code execution can additionally provide feedback and complement the natural language aspect of the verification process. This is consistent with findings in previous works that code-based solutions are better than natural language solutions, such as PAL [1] and PoT [2].
>
> To further demonstrate our point, we provide two examples of natural language failing to find the mistake in the solution, which we put in an anonymous link [3].
>
> As shown in the examples, the natural language verifications in the examples above are weak and inaccurate, failing to find the errors in the solutions. We intend to add these examples to the appendix of the paper.
>
> We hope that the revisions outlined above adequately address your concerns. Thank you once again for pointing out the oversights in our paper. If you have further questions, please let us know.
>
>
> [1] Luyu Gao, Aman Madaan, Shuyan Zhou, Uri Alon, Pengfei Liu, Yiming Yang, Jamie Callan, and Graham Neubig. PAL: Program-aided language models. In International Conference on Machine Learning, 2023.
>
> [2] Wenhu Chen, Xueguang Ma, Xinyi Wang, and William W. Cohen. Program of Thoughts Prompting: Disentangling Computation from Reasoning for Numerical Reasoning Tasks. arXiv preprint arXiv:2211.12588, 2023.
>
> [3] https://anonymous.4open.science/r/CSV_rebuttal_nl_verify-35E1/README.md
>
> *Hope our above responses are helpful to address your concerns. If you have further questions, please let us know. Thanks!*

---

> > ### Comment · Reviewer_iedh · 2023-11-22
> >
> > Thank you for the response, and for all of the revisions – especially the ablations which certainly strengthen the paper and clarify where the benefits are. Overall, I think the results are solid and I support acceptance – this paper proposes two nice solid straightforward and widely-applicable ideas, and finds reasonable strong benefits.
> >
> > The revised results, to summarize myself are:
> > * A gain from adding CSV on top of GPT4-Code of 69.69 -> 73.54 (+3.85)
> > * A gain from adding CSV on top of GPT4-Code + majority voting of 79.88 -> 83.54 (+3.66)
> > * A gain from the proposed modified voting scheme instead of the majority voting scheme of 83.54 -> 84.32 (+0.78)
> > * The missing rows have been added for the results on the other two datasets (tables 2-3) along with the majority voting ablation. Those results look good – nearly everything is >90% accuracy so of course the absolute size of the benefits are smaller but still reasonable.
> >
> > The fixes and clarifications to Fig 6 look good to me.
> >
> > Thank you!

---

> > > ### Comment · Reviewer_iedh · 2023-11-22
> > >
> > > As a follow up note – I've raised my review. Thank you again for the discussion.

---

> > > > ### Author Response · Authors · 2023-11-22
> > > > **Thanks for your response.**
> > > >
> > > > Thank you for acknowledging our rebuttal and efforts.

---

### Author Response · Authors · 2023-11-20
**Paper Update Summary**

Thanks to all the chairs and reviewers for constructive suggestions and comments, we summarize the revised paper details.

1. GPT4-Code + Voting (no CSV) and GPT4-Code + CSV + Voting (no VW-voting) are added in Table 1. (**Reviewer iedh**)
2. Improvement from GPT4-Code + CSV + Voting and GPT4-Code + CSV + VW-Voting are adjusted to be against GPT4-Code + Voting. (**Reviewer iedh, Reviewer qqMb**)
3. GPT4-Code + CSV is added to Table 2 and the inconsistencies in Tables 1, 2, and 3 are fixed. (**Reviewer iedh**)
4. GPT4-Code + CSV + VW-Voting is added to Table 3. (**Reviewer iedh, Reviewer qqMb**)
5. Figure 6a is modified to show only the average result. The dataset is added to the caption of Figure 6. (**Reviewer iedh**)
6. The relationship between Code Usage Frequency and accuracy is corrected on Page 8. (**Reviewer iedh**)
7. Explanation about different sizes of red dots is added to the caption of Figure 2. (**Reviewer iedh**)
8. Clarification about the performance of natural language self-verification is added to Page 8. (**Reviewer iedh**)
9. Failure examples of natural language verification are added to Appendix G. (**Reviewer iedh**)
10. Examples of different types of verification code are added to Appendix H. (**Reviewer boig**)
11. The y-axis is adjusted to start from 0 in Figure 2 and Figure 6a. (**Reviewer boig**)
12. The distribution of the consistency between the reasoning process and the verification process is presented in Figure 9, and the related analysis is added to Appendix I. (**Reviewer boig**)
13. The distribution of verification types generated by GPT4-Code on 50 randomly sampled math word
problems from MATH is presented in Figure 10, and the related analysis is added to Appendix J. (**Reviewer boig**)
14. Experiments of open-source CodeLlama models are added to Appendix C. (**Reviewer qqMb, Reviewer UL1M**)
15. Clarification about performance gain and contribution on Abstract and Page 5-6. (**Reviewer UL1M**)

Apart from the issues addressed above, we also revised our paper with regard to consistency and readability.

---

### Meta-Review · Area_Chair_eY4L · 2023-12-11

**Metareview:**

The paper use OpenAI's new GPT4 with Python Interpreter version. They reveal that incorporating code interpreter as an external could increase LLM's performance on math tasks. The more times using the interpreter, the higher performance would have. In addition, by using the self-debug method with the new GPT4 w/ code, they further push the SoTA of MATH dataset to 84.3%.

Reviewers were most concerned about novelty and how this work only used proprietary GPT4 while other open models would not similar benefit from this method. In the end, the authors addressed most reviewer concerns and the majority of reviewers voted to accept. Most importantly, they provided results showing improvements using Llama as well and thus addressing the main concern raised by the remaining dissenting reviewer that the results must use GPT4.

**Justification For Why Not Higher Score:**

still the main results are about GPT-4

**Justification For Why Not Lower Score:**

reviewer majority

---

### Decision · Program_Chairs · 2024-01-16

Accept (poster)